

# Boundary criticality of complex conformal field theory: A case study in the non-Hermitian 5-state Potts model

**Yin Tang[1,2]⋆, Qianyu Liu[1], Qicheng Tang[3] and Wei Zhu[1]†**

**1** Department of Physics, School of Science, Westlake University, Hangzhou 310030, China
**2** School of Physics, Zhejiang University, Hangzhou 310058, China
**3** School of Physics, Georgia Institute of Technology, Atlanta 30332, USA

⋆ tangyin@westlake.edu.cn , † zhuwei@westlake.edu.cn

## Abstract

Conformal fields with boundaries give rise to rich critical phenomena that can reveal information about the underlying conformality. While most existing studies focus on Hermitian systems, here we explore boundary critical phenomena in a non-Hermitian quantum 5-state Potts model which exhibits complex conformality in the bulk. We identify free, fixed and mixed conformal boundary conditions and observe the conformal tower structure of energy spectra, supporting the emergence of conformal boundary criticality. We also studied the duality relation between different conformal boundary conditions under the Kramers-Wannier transformation. These findings should facilitate a comprehensive understanding for complex CFTs and stimulate further exploration on the boundary critical phenomena within non-Hermitian strongly-correlated systems.

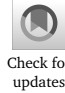

# 1 Introduction

The emergence of conformal invariance offers strong advances in the study of quantum critical phenomena [1–3]. It has been well established that the long-wavelength limit of continuous phase transitions of many lattice models is consistently described by the corresponding conformal field theories (CFTs). Given that the bulk conformality is already very rich and important, the associated boundary critical behavior arising from imposing boundary conditions on critical systems is even more abundant and holds the potential to elucidate a wide range of physical phenomena, ranging from surface critical phenomena in statistical models [4] to quantum transport through impurities [5]. Just like the bulk CFTs, these boundary critical behaviors are believed to fall into corresponding universality classes that are described by boundary CFTs (BCFTs) [6,7].

For a given CFT, there are many possible ways to impose boundary conditions (b.c.s) onto it. In general, all physical boundary conditions are believed to flow to certain boundary fixed points, known as *conformal boundary conditions*. So far, for rational CFTs, the classification of conformal boundary states has achieved great success by using the Verlinde formula, which establishes the correspondence between conformal boundary states and boundary CFT operators [8–11]. In contrast, for generic irrational cases, the analysis of boundary conformal field theory remains challenging both analytically and numerically, and only a few results based mainly on numerical studies are available [12]. However, this difficulty can be partially overcome in models endowed with an enlarged algebraic structure. In particular, in the two-dimensional Potts and $O(n)$ loop models, the presence of the Temperley-Lieb (TL) algebra and its boundary extensions [13–18] allows one to resolve the boundary conformal spectrum for generic $Q < 4$ cases [19–23], even though most of these theories are non-unitary and irrational.

Meanwhile, even when the conformal b.c.s are completely identified in some of the simplest scenarios (such as minimal models), it is still highly non-trivial to align specific boundary states with concrete lattice models, since there is no general principle to ensure the emergence of specific boundary criticality in the low-energy limit for a given boundary interaction term. Notably, the presence of boundaries not only causes local modifications of the correlations, but also leads to a global change in the Hilbert space of CFTs, selecting allowed operators through appropriate boundary conditions and modular invariance [8, 24, 25]. This provides an approach to verify realized conformal boundary states in lattice models by studying the operator content.

This work addresses a new class of irrational CFT, dubbed the *complex CFT,* and their conformal b.c.s. Complex CFT is a branch of conformal invariant theory that greatly violates unitarity, while being intrinsically different from conventional non-unitary CFTs with real conformal data and real central charge (see detailed discussion in [26]). This kind of CFT leads to many interesting physical phenomena. For example, when the unitarity-breaking term is not too strong, the physical system defined in the real parameter space could exhibit approximate conformal symmetry, which is conjectured to be the origin of walking renormalization

group (RG) flows in gauge theories [27–31] and certain weakly first-order phase transitions in statistical physics [26, 32–35]. Very recently, the existence of complex CFTs has been numerically verified in 2-dimensional classical and 1+1-dimensional quantum 5-state Potts models, where the emergent complex Virasoro symmetry and underlying conformal data are revealed explicitly [36, 37].

Moreover, studying complex CFT presents several formidable difficulties and remains a less explored area, leaving many open questions waiting to be investigated. A question of particular interest is the generalization of (non-)unitary BCFT to its complex counterpart. On the one hand, as the complex CFT is irrational, the well-established Verlinde formula [38] does not provide a systematic approach to construct the BCFT from the fusion rules of bulk operators. On the other hand, the boundary conditions of complex CFT are generally complex, for which the stability under RG flows is not known and fully understood. Under these grounds, we try to address the following questions

1. Are there conformal b.c.s, i.e., conformal fixed points under boundary RG flows, for the complex CFT?

2. If these fixed points exist, what are the typical features of boundary conformal invariance in the complex CFT?

Here, in this paper we explore these questions and answer them in the affirmative. We initiate the numerical study on boundary complex CFTs based on the 1+1-dimensional non-Hermitian quantum 5-state Potts model with free, fixed, and mixed boundaries. The emergent boundary complex fixed points are evidenced through the conformal spectrum extracted via an extrapolation of rescaled energy gaps, which is a typical feature for (non-)unitary BCFTs. Using algebraic methods, it is possible to construct the annulus partition function for the $Q < 4$ state Potts model, and some of these results can be analytically continued to the generic $Q$ regime, providing a partial theoretical framework for the complex boundary conditions. For the cases of free b.c. and fixed b.c., we numerically verify the correspondence between the lowest eigenspectrum and Virasoro characters and confirm the duality between free and fixed boundary fixed points. For mixed b.c., we observe emergent complex conformal tower structures in the thermodynamic limit, while a complete determination of all of these Virasoro characters requires further theoretical insight. These observations indicate that the existing knowledge of ordinary BCFT can be (partially) extended to the boundary critical behaviors of complex CFT, but a complete characterization of them requires new tools for complex conformal b.c.s.

This paper is organized as follows. In Sec. 2, we review the known results of the Potts complex CFT, the well-established BCFT formalism for unitary CFTs and the boundary conformal spectrum constructed from blob algebra. We then introduce the non-Hermitian 5-state Potts lattice model and its possible conformal b.c.s in Sec. 3. The main results are presented in Sec. 4, including a detailed analysis of boundary fixed points and related conformal operator content. The duality condition among different b.c.s is also discussed. In Sec. 5, we summarize this work and discuss some open questions for future studies.

## 2 Review of background

### 2.1 Complex CFT and the Potts realization

A generic 2D CFT is characterized by its conformal data, which includes the central charge, the scaling dimensions of local primary operators, and the operator product coefficients among them [1, 3]. The central charge, denoted by a single number $c$, plays a crucial role in the

conformal algebra, which can be decomposed into a direct product of two independent copies of the Virasoro algebra, $\mathcal{V} \otimes \bar{\mathcal{V}}$, generated by $L_n$ and $\bar{L}n$, satisfying

$$
\begin{aligned}
[L_m, L_n] &= (m-n)L_{m+n} + \frac{c}{12}n(n^2-1)\delta_{m+n,0}\,, \\
[\bar{L}_m, \bar{L}_n] &= (m-n)\bar{L}_{m+n} + \frac{c}{12}n(n^2-1)\delta_{m+n,0}\,, \\
[L_m, \bar{L}_n] &= 0\,.
\end{aligned}
\tag{1}
$$

Moreover, each conformal multiplet, including a primary field and all its descendants, constitutes an irreducible representation of the underlying conformal algebra[1] specified by the central charge $c$ [39]. Since unitarity is generally considered essential for quantum theories, most previous studies have focused on CFTs with real $c$ [39]. Nonetheless, the representation theory of the Virasoro algebra with complex $c$ remains mathematically consistent [40].

By extending the definition of conformal data from the real to the complex domain, one can define a so-called complex CFT, which is distinct from ordinary unitary or non-unitary CFTs.[2] Formally, a CFT is unitary if it contains no negative-norm states [3]. If some states have negative norm, the corresponding CFT is non-unitary. Non-unitary CFTs may have negative central charge, negative scaling dimensions, or other unusual properties. A canonical example is the Lee-Yang edge singularity, with central charge $c = -22/5$ and a single nontrivial primary field with $\Delta = -2/5$ besides the identity operator.

Compared with ordinary (non-)unitary CFTs, complex CFTs have several notable features. First, the conformal data (scaling dimensions, OPE coefficients) and the central charge are all complex numbers. Second, in complex CFT, the left state is not the Hermitian conjugate of the right state: $_L\langle \phi | \neq (|\phi\rangle_R)^\dagger$. We use subscripts $L$ and $R$ to distinguish left and right eigenstates, which satisfy biorthogonal relations under radial quantization:

$$
|\phi\rangle_R = \lim_{r\to 0} \hat{\phi}(r,0)|0\rangle_R\,, \qquad {}_L\langle\phi| = \lim_{r\to\infty} r^{2\Delta_\phi} \, {}_L\langle 0|\hat{\phi}(r,0)\,.
\tag{2}
$$

Third, as the central charge is associated with the anomaly of the energy-momentum tensor, it appears in the Hamiltonian and controls the finite-size scaling of the ground-state energy, making it observable [41,42]. In particular, the CFT Hamiltonian (the generator of time translations) defined on a cylinder of circumference $L$—corresponding to a finite-size system of length $L$—is the dilation operator $D = L_0 + \bar{L}_0$ on the complex plane, connected by the conformal mapping $z \to \frac{L}{2\pi}\ln z$, yielding

$$
H = \frac{2\pi}{L}\left(L_0 + \bar{L}_0 - \frac{c}{12}\right).
\tag{3}
$$

Since $c$ is complex in a complex CFT, the Hamiltonian is generally non-Hermitian, implying that complex fixed points must be sought in non-Hermitian models. The Potts model has been proposed as a candidate platform for realizing such complex CFTs, and this conjecture has recently been validated in a non-Hermitian 5-state Potts model [36,37].

Here we briefly recall the two dimensional critical Potts CFT, which can be equivalently formulated through Coulomb gas construction [45] that connects the Potts component $Q$ and the Kac parameter $m$ as

$$
Q = 4\cos^2\left(\frac{\pi}{m+1}\right).
\tag{4}
$$

---

[1]This statement does not hold for some special cases, such as logarithmic CFTs.

[2]Here we distinguish non-unitary CFT from complex CFT: non-unitary CFTs contain real but negative-norm states, whereas complex CFTs have complex conformal data.

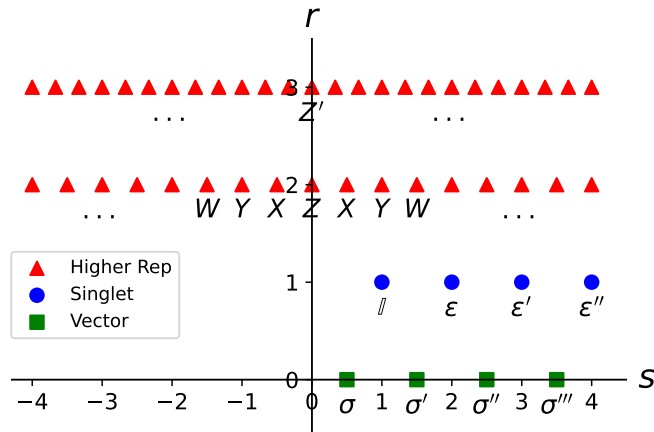

Figure 1: Kac index within $0 \leq r \leq 3$ and $-4 \leq s \leq 4$ for bulk Potts CFT with generic $Q$ [43, 44]. The representations for $S_Q$ singlet operators correspond to the direct product of two irreducible Verma module $V_{1,n} \otimes \bar{V}_{1,n}$ ($n \in \mathbb{N}^*$), marked by blue circles. The $S_Q$ vector fields are also diagonal with Kac index $(0, m + 1/2)$ ($m \in \mathbb{N}$), marked by green squares, while each module is non-degenerate in this situation since $r = 0$. The primary fields with $S_Q$ higher representation correspond to logarithmic operator pairs combining Verma modules with opposite $s$: $V_{r,n/r} \otimes \bar{V}_{r,-n/r}$ or $V_{r,-n/r} \otimes \bar{V}_{r,n/r}$ ($r, n \in \mathbb{N}$ and $r \geq 2$).

The central charge and scaling dimension for each representation could be expressed through the Kac formula

$$c = 1 - \frac{6}{m(m+1)},$$
$$h_{r,s} = \frac{[(m+1)r - ms]^2 - 1}{4m(m+1)}.$$

(5)

This formalism was originally proposed for integer $m$, and recently has been extended to generic $Q$ [35, 43, 44, 46], giving the representation of torus partition function as[3] [44]

$$S_Q = \bigoplus_{s \in \mathbb{N}^*} \mathcal{R}_{(1,s)} \otimes [] \oplus \bigoplus_{s \in \mathbb{N} + \frac{1}{2}} \mathcal{W}_{(0,s)} \otimes [1] \oplus \bigoplus_{r \in \mathbb{N}+2} \bigoplus_{s \in \frac{1}{r}\mathbb{Z}} \mathcal{W}_{(r,s)} \otimes \Xi_{(r,s)}.$$

(6)

Where $[]$, $[1]$ and $\Xi_{(r,s)}$ denote representations of global symmetry $S_Q$, and $\mathcal{R}_{(1,s)}$ and $\mathcal{W}_{(r,s)}$ label Virasoro representations. See an illustration for the diagram of allowed Kac index in Fig. 1. The partition function for the former two kinds of field is always diagonal, while the last field combines $(h_{r,s}, \bar{h}_{r,-s})$ and $(h_{r,-s}, \bar{h}_{r,s})$ forming indecomposable while not fully reducible representation, leading to rank-two Jordan cells under $L_0$ and $\bar{L}_0$. This establishes the fact that the Potts CFT is logarithmic for generic $Q$ [47–49]. Meanwhile, the existence of exact null states for $S_Q$ singlet fields $(h_{1,s}, \bar{h}_{1,s})$ with $s \in \mathbb{N}^*$ imposes strong constraint on the structure of operator algebra, and the associated three-point OPE coefficient and four-point conformal block could be analytically bootstrapped with numerical confirmation [37, 47, 50–55].

Interestingly, when $Q > 4$, from Eq. (4) we have $\cos^2\left(\frac{\pi}{m+1}\right) > 1$, which has no real solution for the parameter $m$. However, it was argued that the Coulomb Gas partition function remains valid after an analytical continuation with complexified $m$ [35]. Consequently, all conformal

---

[3]Note that $Q = 2, 3$ lead to unitary minimal models with finite primary fields, outside of the scope of this partition function.

data, including the central charge, scaling dimensions, and OPE coefficients, are analytically continued into the complex plane. The norm of most CFT states is no longer real simultaneously and the reflection positivity of correlators is not applicable, which strongly conflicts with unitarity. These theories are classified as a novel branch of theories called complex CFTs.

For the Potts case, this means that the existence of real fixed points persists until $Q = 4$, after which real critical points disappear while turning into complex critical points [35]. Since these complex CFTs could only emerge in non-Hermitian models, the phase transition point within the original Potts model could only approach the true RG fixed points, while never reaching it. Thus the nature of phase transitions of 2D Potts model change from continuous to discontinuous at $Q = 4$. As the complex fixed points moving away from the real axis with increasing $Q$, the correlation length at the phase transition points gradually decreases. Importantly, when $Q$ is slightly above 4, a pair of complex fixed points reside extremely close to the real axis, therefore the RG flows between these fixed points greatly slow down and the phase transitions become weakly first-order exhibiting pseudo universal scaling behavior across a large length scale.

## 2.2 Two-dimensional rational BCFT

In this section, we revisit the basic ingredient for 2D rational BCFT. Soon after the birth of two dimensional CFT, Cardy formulated the essential criterion for the conformal b.c.s, stating that the off-diagonal component of the stress tensor, either parallel or perpendicular to the boundary, should vanish [4, 25]. When the boundary aligns with the time axis, this implies the absence of momentum flow across the boundary [7]. Under the RG, any uniform b.c. is expected to flow into a conformally invariant fixed point. Additionally, given a bulk CFT, there may be multiple conformal b.c.s, that are distinctively characterized by their boundary operator content.

For 2D CFT defined in the upper half-plane, the boundary corresponds to the real axis. The conformal b.c. suggests that the energy-momentum tensors fulfill $T(z) = \overline{T}(\overline{z})$ when $z$ lies on the real axis. Consequently, the correlators of $\overline{T}$ are analytically continued from those of $T$ into the lower half-plane. The conformal Ward identity takes the form:

$$\langle T(z)\Pi_j \varphi_j(z_j, \overline{z}_j)\rangle = \sum_j \left( \frac{h_j}{(z - z_j)^2} + \frac{1}{z - z_j}\partial_{z_j} + \frac{\overline{h}_j}{(\overline{z} - \overline{z}_j)^2} + \frac{1}{\overline{z} - \overline{z}_j}\partial_{\overline{z}_j} \right) \langle \Pi_j \varphi_j(z_j, \overline{z}_j)\rangle. \quad (7)$$

In radial quantization, to ensure equivalence among Hilbert spaces defined on different manifolds, one selects semicircles centered at a boundary point, conventionally the origin. Leveraging the conformal b.c., the dilation operator can be expressed as:

$$D = L_0 = \frac{1}{2\pi i}\oint_C zT(z)dz, \quad (8)$$

where $C$ denotes a full circle around the origin. Notably, there is now only a single copy of the Virasoro algebra, due to the fact that conformal mappings preserving the real axis correspond to real analytic functions.

Consideration of the partition function on the torus constrains the bulk operator content through modular invariance. Similarly, consistency on an annulus helps in classifying both permissible b.c.s and boundary operator content. Imagine a CFT on an annulus formed by a rectangle of unit width and height $\delta$, with b.c.s $a$ and $b$ on either edges. The partition function with these b.c.s is denoted as $Z_{ab}(\delta)$.

One approach to compute $Z_{ab}(\delta)$ involves considering the CFT on an infinitely long strip of unit width, conformally related to the upper half-plane by the conformal mapping $z \to (\frac{1}{\pi})\ln z$.

The Hamiltonian equals the generator of infinitesimal translations along the strip, given by:

$$H_{ab} = \pi D - \frac{\pi c}{24} = \pi L_0 - \frac{\pi c}{24}.$$
(9)

For the annulus, we have:

$$Z_{ab}(\delta) = \mathrm{Tr}\, e^{-\delta H_{ab}} = \mathrm{Tr}\, q^{L_0 - \frac{c}{24}},$$
(10)

where $q \equiv e^{-\pi \delta}$. This can be decomposed into characters:

$$Z_{ab}(\delta) = \sum_h n_{ab}^h \chi_h(q),$$
(11)

with the non-negative integers $n_{ab}^h$. Evaluating the coefficients $n_{ab}^h$ fully specifies the operator content for corresponding boundary conformal fixed point, which has mainly been accomplished in rational CFTs.

Alternatively, the annulus partition function can be interpreted, up to an overall rescaling, as the path integral for a CFT on a circle of unit circumference, being propagated for imaginary time $\delta^{-1}$. From this perspective:

$$Z_{ab}(\delta) = \langle a | e^{-H/\delta} | b \rangle.$$
(12)

Since the conformal b.c. requires $L_n = \overline{L}_{-n}$, the solution space for boundary states $|B\rangle$ must reside in the subspace satisfying $L_n |B\rangle = \overline{L}_{-n} |B\rangle$. For diagonal CFTs, a special series of boundary states could be constructed systematically through

$$|h\rangle\rangle = \sum_{N=0}^{\infty} \sum_{j=1}^{d_h(N)} |h, N; j\rangle \otimes |\overline{h, N; j}\rangle,$$
(13)

called Ishibashi states. Then, the overlap between different Ishibashi states is given by

$$\langle\langle h | e^{-\frac{2\pi}{\delta}(L_0 + \overline{L}_0 - \frac{c}{12})} | h'\rangle\rangle = \delta_{hh'} \chi_h(e^{-\frac{4\pi}{\delta}}).$$
(14)

The Cardy states correspond to physical boundary states fulfilling Eq.(11), which are linear combinations of the Ishibashi states:

$$|a\rangle = \sum_h \langle\langle h | a \rangle |h\rangle\rangle,$$
(15)

where $|a\rangle$ and $|h\rangle\rangle$ represent conformal boundary states associated with this b.c. and Ishibashi states linked to boundary primary operator $h$ respectively. Equating the two expression for the annlus partition function, the Cardy conditions are derived as

$$n_{ab}^h = \sum_{h'} S_{hh'} \langle a | h'\rangle\rangle \langle\langle h' | b\rangle,$$

$$\langle a | h'\rangle\rangle \langle\langle h' | b\rangle = \sum_h S_{h'h} n_{ab}^h,$$
(16)

where $S$ is the modular matrix, which characterizes the transformation among different characters under modular transformation $\delta \to -1/\delta$: $\chi_i(q) = \sum_j S_{ij} \chi(\tilde{q})$, with ($\tilde{q} = e^{-\pi/\delta}$) and $\chi_i(q)$ being the conformal character of the corresponding irreducible module labeled by $i$. These requirements impose strong restriction on the structure of permissible boundary states.

It can be demonstrated that the elements $S_{hh'}$ of the modular transformation matrix $S$ also appear in the Verlinde formula [38], derived from considering the consistency of the CFT on a torus. This states that the right-hand side of:

$$n_{h'h''}^h = \sum_\ell \frac{S_{h\ell} S_{\ell h'} S_{\ell h''}}{S_{0\ell}},$$
(17)

is equal to the fusion algebra coefficient $n^h_{h'h''}$. Since these are non-negative integers, the consistency of the above boundary states ansatz is confirmed. At least for diagonal models, there is a bijection between the allowed primary fields in the bulk CFT and the permissible conformally invariant b.c.s.

The construction of Cardy state is helpful, since it shows how boundary primary fields are connected with the modular invariance in the bulk CFT. The aforementioned formalism, including the construction of Ishibashi and Cardy states, applies only to rational BCFTs, where the number of primary fields is finite and the Verlinde formula is well defined. In contrast, for irrational CFTs, the systematic construction of boundary states remains largely unknown. An obvious obstruction is that the number of primaries is infinite, and evaluating the modular $S$-matrix becomes impossible in generic cases. Nevertheless, in the irrational regime of the Potts CFT with $\text{Re}(Q) < 4$, analytic progress can be achieved for a specific class of boundary conditions, in which the boundary spins are restricted to $Q_1 < Q$ possible states. In this case, the boundary Temperley-Lieb algebra can be employed to construct the corresponding annulus partition functions, which exhibit analytic continuation properties in $Q$ analogous to those of the bulk theory. For this reason, we explore properties of the boundary ingredients—the boundary conformal spectrum and the correspondence between boundary fields and Virasoro representations—in the context of the Potts complex CFT. To this end, we perform a lattice study of the above boundary conditions focusing on $Q = 5$ and examine the applicability of the analytic continuation approach in complex BCFTs.

## 2.3 Review on the two-boundary annulus partition functions in the Potts CFT

The construction of annulus partition functions in the two-dimensional Potts and $O(n)$ models is most conveniently formulated within the framework of TL algebras and their boundary extensions [14, 18]. See a more detailed review in [56]. In the bulk, the dense loop representation of the Potts model is governed by the TL algebra $\text{TL}_N(x)$ with generators $\{e_i\}$ satisfying $e_i^2 = xe_i$, $e_i e_{i\pm1} e_i = e_i$, and $e_i e_j = e_j e_i$ for $|i - j| > 1$, where the loop fugacity is $x = q + q^{-1} = 2\cos\gamma$.

**One-boundary case:** For systems with a single boundary, an additional idempotent generator $b$ is introduced, leading to the so-called blob algebra $\mathcal{B}_N(x, y)$ [14,15] that extends $\text{TL}_N$ by marking loops that touch the boundary. A loop carrying the blob receives a modified fugacity $y$ instead of the bulk weight $x$. The blob algebra satisfies the additional relations:

$$b^2 = b, \qquad e_1 b e_1 = y e_1, \qquad e_i b = b e_i, \qquad \text{for} \quad i \geq 2. \tag{18}$$

The representation theory of $\mathcal{B}_N$ classifies two conformal boundary conditions: the blobbed and unblobbed sectors, corresponding respectively to whether the outermost non-contractible loop is allowed or forbidden to touch the boundary [19]. In the scaling limit, the boundary scaling dimensions are given by the Kac weights $h_{r,s} = \frac{[(m+1)r - ms]^2 - 1}{4m(m+1)}$ with central charge $c = 1 - \frac{6}{m(m+1)}$, and the spectrum generating functions for the two sectors are:

$$Z_L^*(r) = \frac{q^{h_{r,r+L} - c/24}}{P(q)} \quad \text{(blobbed sector)}, \tag{19}$$

$$Z_L(r) = \frac{q^{h_{r,r-L} - c/24}}{P(q)} \quad \text{(unblobbed sector)}, \tag{20}$$

where $P(q) = \prod_{n=1}^{\infty}(1 - q^n)$. The full annulus partition function takes the form

$$Z = q^{-c/24}\left[\sum_{j=0}^{\infty} D_{2j}^* \frac{q^{h_{r,r+2j}}}{P(q)} - \sum_{j=1}^{\infty} D_{2j} \frac{q^{h_{r,r-2j}}}{P(q)}\right], \tag{21}$$

where the amplitudes $D_L^*$ and $D_L$ are given by:

$$D_L^* = \frac{\sinh(L\alpha + \beta)}{\sinh\beta}\,, \tag{22}$$

$$D_L = \frac{\sinh(L\alpha - \beta)}{\sinh(-\beta)}\,, \tag{23}$$

with parametrization $l = 2\cosh\alpha$, $m = \frac{\sinh(\alpha+\beta)}{\sinh\beta}$. This framework provides an exact analytic continuation in the loop fugacity $x$ or equivalently in the Potts variable $Q = x^2$, and establishes connections with RSOS models when $q = e^{i\pi/(p+1)}$ with $p$ integer [19].

**Two-boundary case:** For an annulus with two boundaries, the relevant algebraic structure is the two-boundary Temperley–Lieb (2BTL) algebra [18, 21] generated by $\{e_i\}$ together with two idempotent boundary operators $b_1$ and $b_2$. These satisfy the defining relations:

$$b_1^2 = b_1\,, \qquad b_2^2 = b_2\,, \tag{24}$$

$$e_1 b_1 e_1 = n_1 e_1\,, \quad e_{N-1} b_2 e_{N-1} = n_2 e_{N-1}\,, \tag{25}$$

$$e_i b_1 = b_1 e_i\,, \quad \text{for } i \geq 2\,, \tag{26}$$

$$e_i b_2 = b_2 e_i\,, \quad \text{for } i \leq N-2\,, \tag{27}$$

together with the additional quotient condition ensuring that loops touching both boundaries receive the correct weight $n_{12}$. Each closed loop acquires one of the weights

$$n \quad \text{(bulk)}, \qquad\qquad n_1 \quad \text{(left boundary)},$$
$$n_2 \quad \text{(right boundary)}, \quad n_{12} \quad \text{(loop touching both boundaries)},$$

and for non-contractible loops:

$$l \quad \text{(bulk)}, \qquad l_1 \quad \text{(left boundary)}, \qquad l_2 \quad \text{(right boundary)}.$$

The parameters can be parametrized as

$$n = 2\cos\gamma\,, \qquad n_1 = \frac{\sin[(r_1+1)\gamma]}{\sin(r_1\gamma)}\,, \qquad n_2 = \frac{\sin[(r_2+1)\gamma]}{\sin(r_2\gamma)}\,, \tag{28}$$

$$n_{12} = \frac{\sin\left[\frac{\gamma}{2}(r_1+r_2+1-r_{12})\right]\sin\left[\frac{\gamma}{2}(r_1+r_2+1+r_{12})\right]}{\sin(r_1\gamma)\sin(r_2\gamma)}\,, \tag{29}$$

and for non-contractible loops:

$$l = 2\cos\chi\,, \qquad l_1 = \frac{\sin[(u_1+1)\chi]}{\sin(u_1\chi)}\,, \qquad l_2 = \frac{\sin[(u_2+1)\chi]}{\sin(u_2\chi)}\,, \tag{30}$$

where $(r_1, r_2, r_{12}, u_1, u_2)$ are real parameters labeling conformal families.

Combinatorially, the 2BTL modules are organized according to the number of non-contractible strings $s = 2j$ and the blob status (blobbed or unblobbed) of the leftmost and rightmost strings, giving four sectors $\mathcal{V}_{2j}^{bb}$, $\mathcal{V}_{2j}^{bu}$, $\mathcal{V}_{2j}^{ub}$, $\mathcal{V}_{2j}^{uu}$ for each $j \geq 1$. The transfer matrix constructed from these generators acts in these sectors, and its spectrum reproduces the expected conformal towers [21].

Combining algebraic and Coulomb-gas analyses, one can derive the general two-boundary annulus partition function for the Potts model [21]

$$Z_{\text{Potts}}(q) = Z_{\text{loop}}(q) + (Q_{12} - l_1 l_2) Z_{l_1 l_2}(q) , \tag{31}$$

$$Z_{\text{loop}}(q) = \frac{q^{-c/24}}{P(q)} \Bigg[ \sum_{n \in \mathbb{Z}} q^{h_{r_{12}-2n,r_{12}}} + \sum_{j \geq 1} \sum_{n \geq 0} \Big( D_{2j}^{bb} q^{h_{r_1+r_2-1-2n,r_1+r_2-1+2j}}$$
$$+ D_{2j}^{bu} q^{h_{r_1-r_2-1-2n,r_1-r_2-1+2j}} + D_{2j}^{ub} q^{h_{-r_1+r_2-1-2n,-r_1+r_2-1+2j}}$$
$$+ D_{2j}^{uu} q^{h_{-r_1-r_2-1-2n,-r_1-r_2-1+2j}} \Big) \Bigg] , \tag{32}$$

$$Z_{l_1 l_2}(q) = \sum_{j \geq 1} (-1)^{j-1} \big( K_{2j}^{bb} - K_{2j}^{ub} - K_{2j}^{bu} + K_{2j}^{uu} \big) . \tag{33}$$

Here, $Z_{\text{loop}}$ corresponds to the decomposition of the partition function of the original loop model, while the second term $Z_{l_1 l_2}$ accounts for corrections arising from the inequivalent weight representations between the Potts model and the loop model. The coefficients and characters are defined as

$$K_{2j}^{bb} = \frac{q^{-c/24}}{P(q)} \sum_{n \geq 0} q^{h_{r_1+r_2-1-2n,r_1+r_2-1+2j}} , \tag{34}$$

$$K_{2j}^{bu} = \frac{q^{-c/24}}{P(q)} \sum_{n \geq 0} q^{h_{r_1-r_2-1-2n,r_1-r_2-1+2j}} , \tag{35}$$

$$K_{2j}^{ub} = \frac{q^{-c/24}}{P(q)} \sum_{n \geq 0} q^{h_{-r_1+r_2-1-2n,-r_1+r_2-1+2j}} , \tag{36}$$

$$K_{2j}^{uu} = \frac{q^{-c/24}}{P(q)} \sum_{n \geq 0} q^{h_{-r_1-r_2-1-2n,-r_1-r_2-1+2j}} , \tag{37}$$

and

$$D_{2j}^{bb} = \frac{\sin[(u_1 + u_2 - 1 + 2j)\chi] \sin \chi}{\sin(u_1 \chi) \sin(u_2 \chi)} , \tag{38}$$

$$D_{2j}^{bu} = \frac{\sin[(u_1 - u_2 - 1 + 2j)\chi] \sin \chi}{\sin(u_1 \chi) \sin(-u_2 \chi)} , \tag{39}$$

$$D_{2j}^{ub} = \frac{\sin[(-u_1 + u_2 - 1 + 2j)\chi] \sin \chi}{\sin(-u_1 \chi) \sin(u_2 \chi)} , \tag{40}$$

$$D_{2j}^{uu} = \frac{\sin[(-u_1 - u_2 - 1 + 2j)\chi] \sin \chi}{\sin(-u_1 \chi) \sin(-u_2 \chi)} . \tag{41}$$

This exact result provides a unified expression valid for generic real $Q = n^2 < 4$, including both rational and irrational regimes.

At special rational points $q = e^{i\pi/(m+1)}$ with integer $m$, the 2BTL representation truncates, and the resulting conformal spectra coincide with those of the $A_m$ RSOS minimal models with corresponding boundary conditions. Away from these points, the same algebraic structure furnishes an analytic continuation to the non-unitary irrational Potts CFT with $\text{Re}(Q) < 4$. The Temperley–Lieb algebra and its boundary extensions thus provide an exact combinatorial and algebraic framework for constructing annulus partition functions and resolving boundary conformal spectra even in the irrational regime.

# 3 Lattice model

## 3.1 Original Potts model

The original $Q$-state quantum Potts model is defined as [57]

$$
\begin{aligned}
H_{\texttt{Potts}} &= H_0(J,h) - g_L \eta_L - g_R \eta_R, \\
H_0(J,h) &= -J \sum_{i=1}^{L-1} \sum_{k=1}^{Q-1} (\sigma_i^\dagger \sigma_{i+1})^k - h \sum_{i=1}^{L} \sum_{k=1}^{Q-1} \tau_i^k.
\end{aligned}
\tag{42}
$$

The Hilbert space is a tensor product of $Q$-dimensional local spaces spanned by $|0,1,\cdots,(Q-1)\rangle$. $H_0$ is the bulk Hamiltonian, and $\eta_{L(R)}$ is the boundary term.

We first focus on the bulk Hamiltonian $H_0$. The Potts spin phase operator is defined as

$$
\hat{\sigma}|n\rangle = \omega^n |n\rangle, \qquad \omega = e^{2\pi i/Q},
\tag{43}
$$

and the spin flip operator is

$$
\hat{\tau}|n\rangle = |(n+1) \bmod Q\rangle.
\tag{44}
$$

Importantly, they satisfy the relations

$$
\sigma_i^Q = \tau_i^Q = 1, \qquad \sigma_i \tau_i = \tau_i \sigma_i \omega.
\tag{45}
$$

$H_0(J,h)$ has two essential properties:

(1) The Hamiltonian is invariant under the $S_Q$ spin permutation symmetry. For example, for $Q=3$, this symmetry is generated by the $Z_3$ symmetry generator $\mathcal{S}$ and charge conjugation $\mathcal{C}$. Their actions do not commute, and the non-Abelian nature of $S_Q$ has interesting consequences. Charge conjugation obeys $\mathcal{C}^2 = 1$ and acts on the operators as

$$
\mathcal{C}\sigma_j \mathcal{C} = \sigma_j^\dagger, \qquad \mathcal{C}\tau_j \mathcal{C} = \tau_j^\dagger.
\tag{46}
$$

Cyclic permutations are generated by $\mathcal{S} = \Pi_j \tau_j$, and they satisfy

$$
\mathcal{S}^\dagger \sigma_j \mathcal{S} = \omega \sigma_j, \qquad \mathcal{S}^\dagger \tau_j \mathcal{S} = \tau_j.
\tag{47}
$$

(2) For general $Q$ and periodic b.c., the Potts model is self-dual at the point $J=h$, under the Kramers-Wannier transformation

$$
\tau_i \to \sigma_i^\dagger \sigma_{i+1}, \qquad \sigma_i^\dagger \sigma_{i+1} \to \tau_{i+1}.
\tag{48}
$$

By tuning the parameter $J/h$, $H_0(J,h)$ exhibits two phases: an ordered phase at $J > h$ that spontaneously breaks $S_Q$ symmetry, and a disordered phase at $J < h$ that preserves $S_Q$ symmetry. The order-disorder transition occurs at $J=h$, with its precise position determined by the Kramers-Wannier duality. It is well established that the phase transition is continuous for $Q \le 4$, but becomes first-order for $Q > 4$ [58–61]. Notably, for $Q$ just above 4, such as $Q=5$, the first-order transition is very weak, characterized by a large correlation length and a small energy gap [62].

Next, we turn to the boundary terms. From the perspective of the Q-state quantum Potts model, different known boundary states are achieved by adding a large boundary field that projects the boundary spin to a specific configuration. For example, in the 3-state Potts CFT, corresponding to the D series of the minimal model $\mathcal{M}_{5,6}$ with central charge $c = \frac{4}{5}$ [63], the free, fixed (A,B or C), and mixed (AB,BC,AC) b.c.s [64–66] have been established as conformal b.c.s that renormalize into the corresponding boundary fixed points of 3-state Potts BCFT [8,

67], where A, B, and C correspond to the three spin directions at each local site, related via $Z_3$ cyclic permutation. The complete classification of Cardy states includes another conformal boundary state $|\phi_{2,2}\rangle$ [67,68], realized as the "new" boundary condition,[4] preserving the $S_3$ permutation symmetry. It can be reached by changing the sign of the boundary transverse fields and polarizing the boundary spins along the direction of this transverse field: $\frac{|A\rangle+|B\rangle+|C\rangle}{\sqrt{3}}$ [66].

## 3.2 Non-Hermitian Potts model

The original Potts model $H_{\text{Potts}}$ in Eq.(42) exhibits second-order ferromagnetic-to-paramagnetic phase transitions for $Q \leq 4$. For $Q > 4$, the phase transition becomes first-order, so in the traditional viewpoint, no critical phenomena are expected. On the other hand, it has been proposed that the $Q > 4$ Potts model could host conformal fixed points in the complex parameter space [35], which has been realized and validated in an extended quantum lattice model [37] and a classical model [36] recently.

In this work, we mainly follow Ref. [37] and consider a $1 + 1$D non-Hermitian 5-state Potts model subjected to different b.c.s:

$$H_{\text{NH-Potts}} = H_0(J,h) + H_1(\lambda) - g_L \eta_L - g_R \eta_R,$$
$$H_1(\lambda) = \lambda \sum_{i=1}^{L-1} \sum_{k_1,k_2=1}^{Q-1} [(\tau_i^{k_1} + \tau_{i+1}^{k_1})(\sigma_i^\dagger \sigma_{i+1})^{k_2} + (\sigma_i^\dagger \sigma_{i+1})^{k_1}(\tau_i^{k_2} + \tau_{i+1}^{k_2})], \tag{49}$$

where $H_1(\lambda)$ is a nearest-neighbor interaction term which preserves the Kramers-Wannier duality and $S_Q$ spin permutation symmetry. By choosing $\lambda$ as a complex number, $H_1(\lambda)$ breaks the hermiticity of the original Potts model. For the bulk Hamiltonian ($g_L = g_R = 0$) with periodic b.c., $H_{\text{NH-Potts}}$ realizes two fixed points in the complex parameter space, which are complex-conjugate partners described by complex CFT [37]. The critical parameters are determined as $J_c = h_c = 1$ and $\lambda_c \approx 0.079 + 0.060i$ for $Q = 5$.

The boundary terms in Eq.(49) introduce different b.c.s through different choices of $\eta$ applied to the ends of the spin chain ('L(R)' labels left (right) boundary site). Inspired by previous conformal b.c.s in 2-, 3-, and 4-state Potts models [8,65,66,69], we consider free, fixed, and mixed b.c.s in this work. As mentioned above, these are also known as blob b.c.s, where the boundary spin can only take $Q_b \leq Q$ values [14]. For $Q_b = Q$, $\eta$ is a trivial null matrix and the boundary is free. For $Q_b = 1$, the boundary spin can only take a fixed value, corresponding to fixed b.c., which can be realized by adding a strong magnetic field that projects the boundary spin to a certain direction. Similarly, for mixed b.c. $1 < Q_b < Q$, $\eta$ forbids several boundary spin values while allowing the remaining spins with equal probability. To ensure that the imposed boundary conditions flow to the corresponding boundary fixed points, we set $g_L = g_R = 20$ in the numerical calculations, which is at least one order of magnitude larger than all coupling constants in the bulk Hamiltonian. In addition, both $g_L$ and $g_R$ should be positive; negative values would correspond to different boundary fixed points. The correspondence between b.c.s and different choices of $\eta$ is summarized in Table 1.

## 3.3 Blob boundary condition and quantum group symmetry

In order to define boundary conditions that preserve the quantum-group symmetry, it is convenient to recall the construction of the so-called blob (or boundary Temperley-Lieb) algebra [19,21]. The bulk Temperley-Lieb algebra is generated by $e_i$ with loop weight $x = 2\cos\gamma$ and $Q = x^2$. The open chain admits a natural $U_q(\mathfrak{sl}_2)$ symmetry, with $q = e^{i\gamma}$. Adding a

---

[4]In particular, this new boundary condition was found by implementing a duality between different boundary conditions, which we will discuss later.

Table 1: Conformal boundary conditions for the non-Hermitian 5-state Potts model. We use {A,B,C,D,E} to label allowable values of boundary spins.

| Boundary condition | $\eta$ |
|---|---|
| free | Null matrix |
| fixed (A) | $\lvert 0\rangle\langle 0\rvert - \sum_{i=1}^{4}\lvert i\rangle\langle i\rvert$ |
| mixed (ABCD) | $\sum_{i=0}^{3}\lvert i\rangle\langle i\rvert - \lvert 4\rangle\langle 4\rvert$ |

boundary is achieved algebraically by introducing a projector $b$, which defines the blob algebra. When the boundary parameter is taken as

$$y = \frac{\sin((r+1)\gamma)}{\sin(r\gamma)}, \qquad r \in \mathbb{Z}_{>0}, \tag{50}$$

the boundary projector $b_r$ can be constructed explicitly by "cabling": one introduces $r-1$ auxiliary (ghost) spin-$\frac{1}{2}$ chains and performs a $q$-symmetrization with the first physical spin. This $q$-symmetrizer projects onto the spin-$r/2$ representation of $U_q(\mathfrak{sl}_2)$, so that the whole boundary Hilbert space remains a finite-dimensional representation of the same quantum group. In this way, the blob boundary condition (50) is fully compatible with the quantum-group symmetry of the bulk Hamiltonian. The corresponding loop/cluster weight on a boundary-touching loop is

$$Q_s = x\,y = \sqrt{Q}\,\frac{\sin((r+1)\gamma)}{\sin(r\gamma)}, \tag{51}$$

which is interpreted as the "effective number" of boundary spin states $Q_s < Q$ in the Potts spin formulation. Importantly, $Q_s$ here is a loop weight, not an integer color multiplicity: the reduction of $Q_s$ arises from projecting the boundary degrees of freedom onto a definite $U_q(\mathfrak{sl}_2)$ representation.

For $Q > 4$, the same relations can be analytically continued by writing $x = 2\cosh\lambda$ (with $\lambda > 0$) and $q = e^\lambda$. Equation (50) then becomes

$$y(r) = \frac{\sinh((r+1)\lambda)}{\sinh(r\lambda)}, \qquad Q_s(r) = x\,y(r) = \sqrt{Q}\,\frac{\sinh((r+1)\lambda)}{\sinh(r\lambda)}. \tag{52}$$

For instance, at $Q = 5$ one has $\lambda = \operatorname{arccosh}(\sqrt{5}/2)$. The $Q_s$ as a function of $r$ for $Q = 5$ is shown in Fig. 2. At integer $r$ the construction (52) still gives a well-defined boundary projector, so that the quantum-group symmetry is formally preserved by analytic continuation.

For $r = 1$ we find $Q_s = 5$, corresponding to the free boundary condition. For $r = 2$ one obtains $Q_s = 4$, which may be interpreted as a "mixed" boundary condition. For larger positive integers $r \geq 3$, one obtains a family of non-integer values $3 < Q_s < 4$, approaching asymptotically

$$Q_s(+\infty) = \sqrt{5}\,e^\lambda \approx 3.618, \tag{53}$$

as $r \to +\infty$, see Fig. 2. Although these values of $Q_s$ cannot literally be realized in the quantum spin model with integer degrees of freedom, they can be implemented naturally in the loop representation by assigning the corresponding fugacity to loops touching the boundary [36].

If one formally allows negative integers $r$, Eq. (52) still yields real positive values of $Q_s$. For example, $r = -2$ gives $Q_s = 1$, which coincides with the usual fixed boundary condition of the Potts CFT. However, for negative or non-integer $r$ the cabling construction of the boundary projector no longer applies—there are no "$r$ ghost spins" to symmetrize—and hence such cases should be regarded merely as an analytic continuation of the blob-algebra parameter. They do not correspond directly to the finite-dimensional $U_q(\mathfrak{sl}_2)$ representations that arise in the original construction.

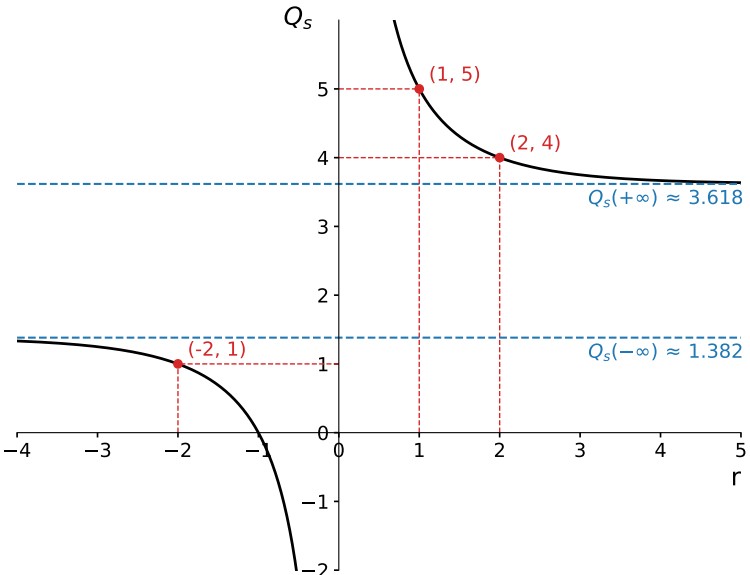

Figure 2: Analytic continuation of the blob-boundary parameter $Q_s(r)$ for $Q = 5$. The curve shows $Q_s = \sqrt{Q} \sinh[(r+1)\lambda]/\sinh(r\lambda)$ with $\lambda = \text{arccosh}(\sqrt{5}/2)$. The special integer points $r = 1, 2, -2$ correspond respectively to the free ($Q_s = 5$), mixed ($Q_s = 4$), and fixed ($Q_s = 1$) boundary conditions. For $r \geq 3$, the boundary fugacity $Q_s$ takes non-integer values between 3 and 4, approaching $Q_s(+\infty) \approx 3.618$ as $r \to +\infty$. The dashed blue lines indicate the asymptotic limits $Q_s(\pm\infty)$.

## 4 Numerical results

### 4.1 Gaplessness and finite-size scaling

We numerically study the non-Hermitian Hamiltonian in Eq.(49) using exact diagonalization (ED) and extract its low-lying eigenvalue spectra for different total system sizes $L$ (from $L = 6$ to $L = 11$). Near the boundary critical point, the energy spectrum is expected to follow the BCFT scaling form [42, 70]:

$$(E_n - E_0) \propto \frac{\pi}{L}\left(h_n - \frac{c}{24}\right) + \delta E_n^{\text{bulk}} + \delta E_n^{\text{boundary}}, \tag{54}$$

where $E_n$ and $E_0$ are the eigenenergies of the $n$-th excited state and the ground state, $h_n$ is the scaling dimension of the corresponding boundary operator, and $c$ is the central charge. $\delta E_n^{\text{bulk}}$ and $\delta E_n^{\text{boundary}}$ are finite-size corrections arising from bulk and boundary irrelevant operators [24, 71–73], whose contributions vanish in the thermodynamic limit. Eq.(54) formally takes the same form as in usual BCFT because it follows from global conformal symmetry. The only difference is that, in complex CFT, conformal data such as the central charge $c$ and scaling dimensions $h_n$ are generally complex. As a result, the eigenenergies $E_n$ have both real and imaginary parts, which are shown separately in Fig. 3. We illustrate the procedure for free-free b.c.s. Both the real and imaginary parts of the raw energy gaps scale to nearly zero in the thermodynamic limit, signaling boundary criticality.

The bulk correction $\delta E_n^{\text{bulk}}$ takes the same form as in the periodic boundary condition case, independent of the boundary condition, as analyzed in [24, 37]. The leading-order correction $\delta E_n^{\text{bulk}} \propto (1/L)^3 + \cdots$ arises from $T^2 + \bar{T}^2$ and $T\bar{T}$. The boundary correction $\delta E_n^{\text{boundary}}$ depends on the operator content of the corresponding boundary fixed points and needs to be treated case by case [73]. For example, for free-free b.c., the leading-order term of

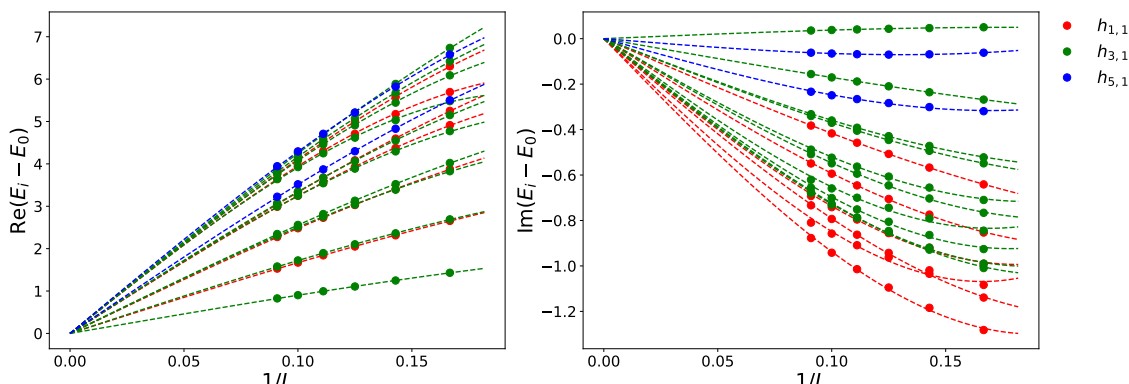

Figure 3: Finite-size scaling of raw energy gaps with free-free boundary condition. We numerically compute several low-lying energy gaps with total system size $L = 6, 7, \cdots, 11$ and fit them through $(E_n - E_0) \propto A/L + B/L^3$. Vanishing energy gaps in the thermodynamic limit signals the boundary criticality. Different colors label states belonging to different conformal multiplets, which will be elucidated in detail below. The extracted speed of light $v = \frac{A}{2\pi} \simeq 2.7205 - 0.6906i$ is close to the periodic case where $v_{\mathrm{PBC}} \simeq 2.8810 - 0.7091i$ has been numerically determined in [37].

$\delta E_n^{\mathrm{boundary}}$ is higher than $(1/L)^4$.[5] Therefore, in Fig. 3, we use the finite-size scaling function $(E_n - E_0) \propto A/L + B/L^3$.

## 4.2 Conformal tower

After confirming the existence of boundary criticality, we extract the scaling dimensions of boundary operators through the rescaled gap:

$$\frac{2(E_n - E_0)}{E_{L_{-2}\mathbb{I}} - E_0} \approx h_n + \frac{\alpha}{L^2} + \cdots. \tag{55}$$

The extrapolation of rescaled energy gaps is shown in Fig. 4. Here, we identify three boundary primary operators with $\mathrm{Re}(h_n) \leq 5$ for the free-free b.c., labeled as $\phi_{1,1}, \phi_{3,1}$, and $\phi_{5,1}$ (see the Kac table in Fig. 1), corresponding to the lowest $S_5$ singlet, standard, and other representation boundary operators. Since this b.c. preserves the full permutation symmetry, only perturbations from $S_5$ singlet boundary operators contribute to the energy spectrum. From our numerical results, the only singlet primary operator with a real part of the scaling dimension lower than 5 is the identity operator, indicating that the leading boundary contribution to $\delta E_n^{\mathrm{boundary}}$ is higher than $1/L^4$. Accordingly, the boundary conformal spectrum can be extracted via an extrapolation of rescaled energy gaps in Eq. (55), and the results are shown in Fig. 4. In the thermodynamic limit, it is expected that all states within the same conformal family share the same imaginary part of the scaling dimension, while their real parts differ by integer values.

Next, we separately discuss the conformal tower structures for different b.c.s.

---

[5]The boundary perturbation can be inspected from the boundary conformal spectrum, which will be presented in the next subsection. For now, we assume this condition holds and will recheck it when the boundary operator content is extracted.

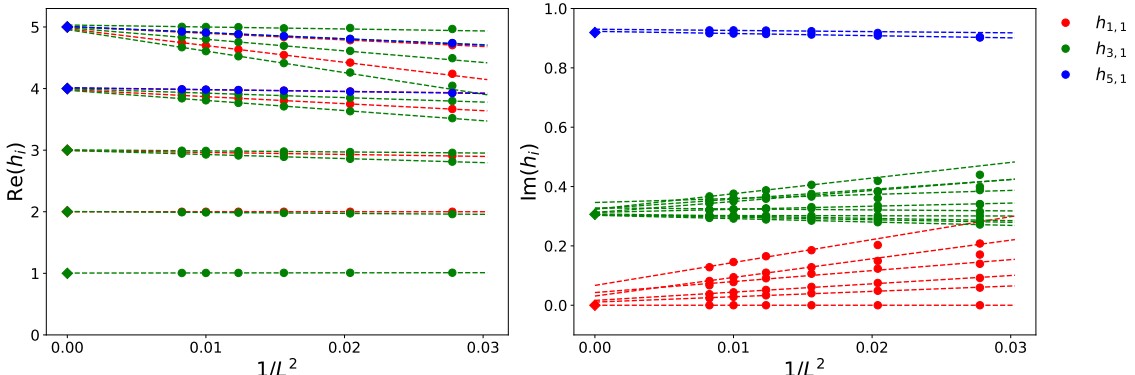

Figure 4: Finite-size scaling of rescaled energy gaps with free-free boundary condition. At each size, we rescale the whole spectrum by setting the first descendant of the identity operator to be $h_{L_{-2}\mathbb{I}} = 2$. Then an extrapolation is performed through $h(L) = h(\infty) + C/L^2$. Up to $\mathrm{Re}(h) \leq 5$ (here $h$ labels the scaling dimension of the boundary field), we classified three conformal multiplets according to their degeneracy and conformal tower structure. The $h_{1,1}$ multiplet labels the conformal family of identity operators, whose lowest field corresponds to the ground state in this case. The $h_{3,1}$ multiplet denotes the most relevant $S_5$ 4-dimensional standard representation operators, corresponding to the scaling dimension of boundary magnetization. Lastly, the representation of the field $h_{5,1}$ under $S_5$ decomposes as a direct sum of irreducible representations with 11-fold degeneracy.

### 4.2.1 Free-free b.c.

We now discuss the boundary conformal towers in this model. The lowest scaling dimensions of boundary operators for free-free b.c. are summarized in Tab. 2 and Fig. 5. The boundary conformal spectrum exhibits many universal features, revealing underlying conformal information. First, within each conformal family, the operator with the lowest real part of the scaling dimension is identified as a primary boundary operator $\hat{\phi}$, with its descendants differing from it by integer values. This is because descendant fields are generated by applying Virasoro generators: $L_{-\nu_1} L_{-\nu_2} \cdots L_{-\nu_m} \hat{\phi}$ with positive integer $\nu$. For BCFT, there is only one copy of the Virasoro algebra, in contrast with bulk criticality, where both holomorphic and anti-holomorphic parts are relevant. Additionally, most scaling dimensions for boundary operators are complex, in contrast to BCFTs with real fixed points. By comparing the scaling dimensions with the highest weights of the corresponding Verma modules, we can align the lowest boundary operators with their representations and Kac indices under the Virasoro algebra. Furthermore, the $rs$-th level descendants contain a null state for the Verma module with integer $r$ and $s$, which is explicitly observed in our numerics. For example, the third-order descendants of the Kac operator $\phi_{3,1}$ have only twofold degeneracy, due to the constraint $f_1 L_{-1}^3 \phi_{3,1} + f_2 L_{-1}L_{-2}\phi_{3,1} + f_3 L_{-3}\phi_{3,1} = 0$ ($f_i$ are constants related to the central charge $c$). Both the theoretical value and degeneracy agree with the numerical results.

Based on our numerical results, the lowest contributions to the cylinder partition function of the $S_5$ complex fixed points with free-free b.c. can be determined as

$$Z_{\text{free,free}} = \chi_{1,1} + 4\chi_{3,1} + 11\chi_{5,1} + \cdots , \tag{56}$$

where the coefficient in front of each Virasoro character represents the degeneracy of the corresponding conformal field, and the subscript denotes its Kac index under the Virasoro algebra. Interestingly, this boundary partition function is also revealed through the boundary entanglement spectrum from recent tensor-network computations [74].

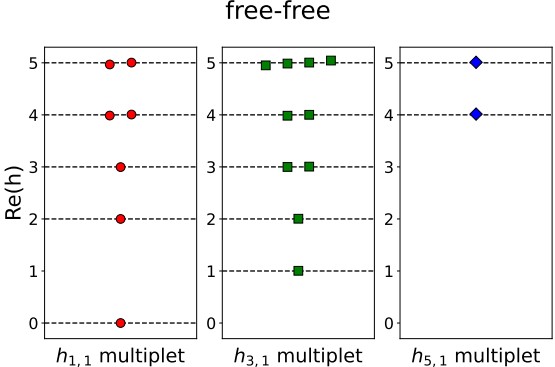

Figure 5: Real parts of boundary conformal spectrum of free-free boundary condition. The solid symbols label numerical results after an extrapolation (see main text), while the dashed lines marked theoretical values from each identified Verma module. Different multiplets are distinguished through their ground states degenercay, originating from different $S_Q$ representations. The imaginary parts belonging to the same conformal multiplet are almost the same, see Tab.2.

We also presented the small $q$ expansions of several Virasoro characters as:

$$
\begin{aligned}
\chi_{1,1} &= q^{-c/24+h_{1,1}}(1 + q^2 + q^3 + 2q^4 + 2q^5 + 4q^6 + \cdots), \\
\chi_{2,1} &= q^{-c/24+h_{2,1}}(1 + q + q^2 + 2q^3 + 3q^4 + 4q^5 + 6q^6 + \cdots), \\
\chi_{3,1} &= q^{-c/24+h_{3,1}}(1 + q + 2q^2 + 2q^3 + 4q^4 + 5q^5 + 8q^6 + \cdots), \\
\chi_{4,1} &= q^{-c/24+h_{4,1}}(1 + q + 2q^2 + 3q^3 + 4q^4 + 6q^5 + 9q^6 + \cdots), \\
\chi_{5,1} &= q^{-c/24+h_{5,1}}(1 + q + 2q^2 + 3q^3 + 5q^4 + 6q^5 + 10q^6 + \cdots),
\end{aligned}
\tag{57}
$$

where the central charge $c$ and scaling dimension $h_{r,s}$ could be evaluated through Eq.(5).

These results can also be understood from the 2BTL annulus partition function. For the free–free b.c. case, the corresponding blob parameters are $r_1 = r_2 = r_{12} = 1$ ($Q = 5$). By substituting these values into the general 2BTL formula Eq. (31), one recovers exactly the decomposition, as shown in Eq. (58). Moreover, the coefficients 1, 4, and 11 in front of the Virasoro characters coincide with the even-order Chebyshev polynomials of the second kind, which matches the well-known result derived from Temperley–Lieb representation theory [75].

$$
Z^{2BTL}_{\text{free,free}} = \chi_{1,1} + 4\chi_{3,1} + 11\chi_{5,1} + 29\chi_{7,1} + 76\chi_{9,1} + \cdots.
\tag{58}
$$

Here we use the superscript "$2BTL$" to distinguish our numerical results from the annulus partition function obtained via analytic continuation in Eq. (31). Moreover, for the free-free case, $Q_{12} - l_1 l_2 = 0$, so in this situation the boundary partition function of the loop model coincides with that of the Potts model.

### 4.2.2 Fixed b.c.

Next, we consider fixed b.c. (A-A, A-B, free-A), in which the boundary spin is polarized into a fixed direction among $Q$ possible values [65,66,69]. We verified the emergent boundary criticality associated with these b.c.s via the vanishing of energy gaps and extracted the underlying operator content through extrapolation, in parallel to the discussion for free-free b.c. Let us consider the finite-size correction within the A-A b.c. as an example. The boundary transverse field breaks the full $S_5$ symmetry into its $S_4$ subgroup. Thus, only $S_4$ singlet operators with one-fold degeneracy modify the scaling form. From the numerical results, the Kac operator

Table 2: Boundary conformal multiplet for free-free boundary condition. The first column presented the representation of different boundary operators $\phi_{r,s}$ specified by Kac index $r,s$. Then the theoretical prediction and numerical results after extrapolation are presented within the second and third column. In the numerical analysis, the two lowest non-degenerate states are normalized to have scaling dimensions 0 and 2, respectively, in order to rescale the entire spectrum. These two states correspond to the identity operator $\phi_{1,1}$ and its first non-null descendant $L_{-2}\phi_{1,1}$. Consequently, their scaling dimensions are exactly 0 and 2 in the numerical results.

| Operators | theoretical | numerical |
|:---:|:---:|:---:|
| $\phi_{1,1}$ | 0 | 0 |
| $L_{-2}\phi_{1,1}$ | 2 | 2 |
| $L_{-3}\phi_{1,1}$ | 3 | $2.9961 + 0.0069i$ |
| $L^2_{-2}\phi_{1,1}$ | 4 | $3.9868 + 0.0287i$ |
| $L_{-4}\phi_{1,1}$ | 4 | $4.0060 + 0.0115i$ |
| $L_{-2}L_{-3}\phi_{1,1}$ | 5 | $4.9681 + 0.0764i$ |
| $L_{-5}\phi_{1,1}$ | 5 | $5.0044 + 0.0351i$ |

| Operators | theoretical | numerical |
|:---:|:---:|:---:|
| $\phi_{3,1}$ | $1 + 0.3063i$ | $1.0037 + 0.3060i$ |
| $L_{-1}\phi_{3,1}$ | $2 + 0.3063i$ | $2.0042 + 0.3045i$ |
| $L^2_{-1}\phi_{3,1}$ | $3 + 0.3063i$ | $2.9990 + 0.3065i$ |
| $L_{-2}\phi_{3,1}$ | $3 + 0.3063i$ | $3.0055 + 0.3033i$ |
| $L^3_{-1}\phi_{3,1}$ | $4 + 0.3063i$ | $3.9836 + 0.3200i$ |
| $L_{-1}L_{-2}\phi_{3,1}$ | $4 + 0.3063i$ | $4.0010 + 0.3090i$ |
| $L^4_{-1}\phi_{3,1}$ | $5 + 0.3063i$ | $4.9512 + 0.3524i$ |
| $L^2_{-1}L_{-2}\phi_{3,1}$ | $5 + 0.3063i$ | $4.9857 + 0.3281i$ |
| $L^2_{-2}\phi_{3,1}$ | $5 + 0.3063i$ | $5.0047 + 0.3142i$ |
| $L_{-4}\phi_{3,1}$ | $5 + 0.3063i$ | $5.0435 + 0.3223i$ |

| Operators | theoretical | numerical |
|:---:|:---:|:---:|
| $\phi_{5,1}$ | $4 + 0.9190i$ | $4.0148 + 0.9216i$ |
| $L_{-1}\phi_{5,1}$ | $5 + 0.9190i$ | $5.0097 + 0.9319i$ |

$\phi_{5,1}$ acquires the $S_4$ 3-dimensional standard representation, while the other primary with dimension smaller than 5 is the identity operator. Therefore, the finite-size correction takes the same form as for free-free b.c., and the extrapolation formula Eq. (55) remains valid in this case. For the A-B or free-A b.c.s, the identity multiplet is not included within the boundary operator content, which is a generic consequence when the b.c.s applied to the two ends are different. This implies that the entire spectrum is shifted by an overall constant corresponding to the lowest scaling dimension. However, the boundary perturbation can still be safely neglected due to the absence of irrelevant boundary singlet operators up to $\text{Re}(h) = 5$. For these two cases, we rescale the spectrum by setting the gap between the lowest singlet field $\psi$ and its first-order descendant to 1 and perform the extrapolation as:

$$\frac{E_n - E_0}{E_{L_{-1}\psi} - E_\psi} \approx h_n + \frac{\alpha}{L^2} + \cdots. \tag{59}$$

Again, we can align each boundary operator with its irreducible representation under the Virasoro algebra based on the existence of null states and theoretical predictions for scaling dimensions. The numerical results are presented in Fig. 6, where the spectrum also displays

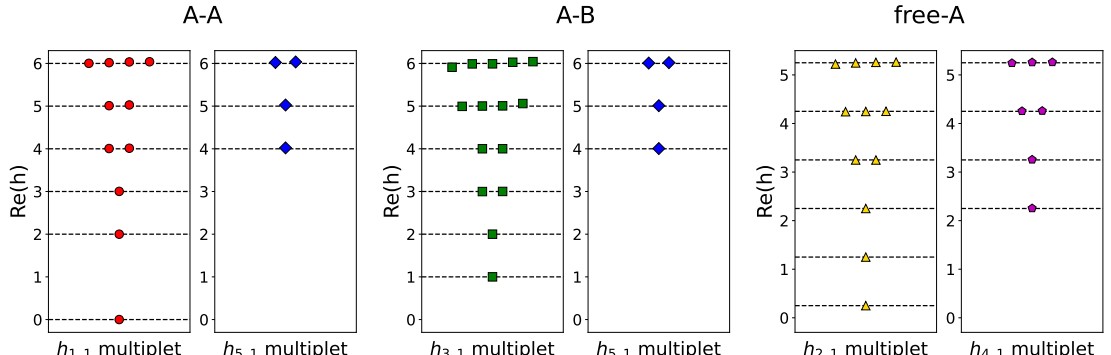

Figure 6: Boundary conformal spectrum of A-A, A-B and free-A boundary conditions. The numerical results are labeled by solid symbols, and the dashed lines represents the theoretical values of scaling dimensions within each Virasoro character.

boundary conformal tower structure, arising from the fusion of different conformal boundary states [8].

For fixed b.c.s, we determine the corresponding cylinder partition functions over Virasoro characters as

$$Z_{\text{A,A}} = \chi_{1,1} + 3\chi_{5,1} + \cdots, \tag{60}$$

$$Z_{\text{A,B}} = \chi_{3,1} + 2\chi_{5,1} + \cdots, \tag{61}$$

$$Z_{\text{free,A}} = \chi_{2,1} + 3\chi_{4,1} + \cdots. \tag{62}$$

These results can also be interpreted within the analytical framework of the 2BTL algebra. In particular, the identification of the free-fixed boundary condition changing operator as $\phi_{2,1}$ was first established by Cardy [76] in the percolation case ($Q = 1$) and subsequently generalized to arbitrary $Q$ within the blob-algebra approach by Jacobsen and Saleur [19], and further analyzed in the two-boundary setting in Ref. [21]. Within this formalism, the fixed–fixed partition function $Z_{A,A}$ follows from Ref. [19], leading to the coefficient $Q-2 = 3$ for the first nontrivial term, in agreement with our numerical observation in Eq. (60). Explicitly, the two-boundary construction (31) gives

$$Z_{A,A}^{2BTL} = \chi_{1,1} + 3\,\chi_{5,1} + 5\,\chi_{7,1} + 16\,\chi_{9,1} \tag{63}$$

$$\textcolor{red}{+\chi_{1,3} + 3\,\chi_{5,3} + 5\,\chi_{7,3} + \cdots,} \tag{64}$$

$$Z_{A,B}^{2BTL} = \chi_{3,1} + 2\,\chi_{5,1} + 6\,\chi_{7,1} + 15\,\chi_{9,1} \tag{65}$$

$$\textcolor{red}{+\chi_{3,3} + 2\,\chi_{5,3} + 6\,\chi_{7,3} + \cdots.} \tag{66}$$

Although several of these Virasoro characters and their degeneracies agree quantitatively with our numerical findings, the full analytical expressions contain additional contributions (marked in red). We conjecture two possible reasons for this. First, under the fixed boundary conditions considered above, one needs to analytically continue the parameter $r$ to $-2$, while in the original 2BTL construction $r$ can only take positive integer values. Second, for $Q > 4$, where the Potts model exhibits a complex CFT, the loop representation of the annulus partition function requires including extra contributions, similar to the bulk case [36]. Such an extension of the 2BTL construction may provide a complete reproduction of our numerical results and clarify the modification of the partition-function structure in the complex CFT regime.

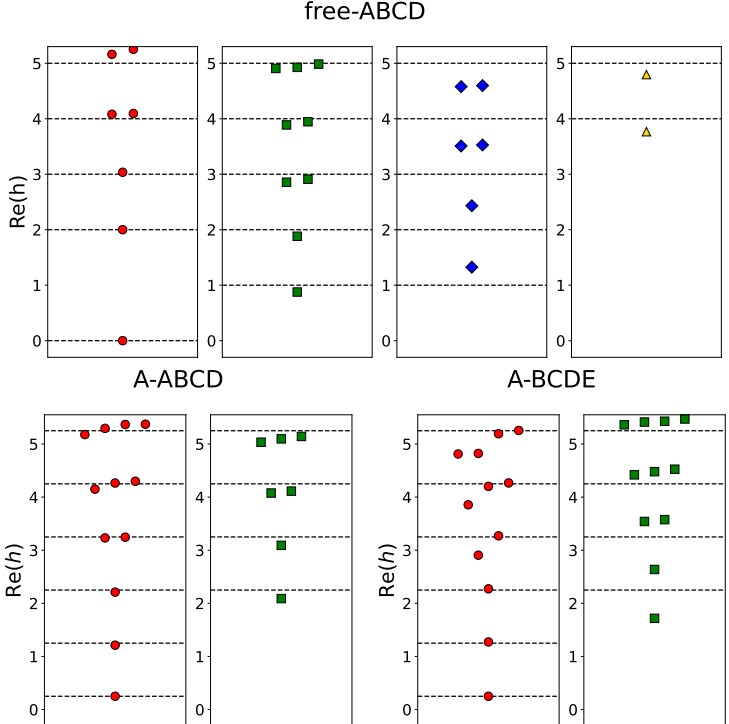

Figure 7: Boundary conformal spectrum of free-ABCD, A-ABCD, and A-BCDE boundary conditions. Solid symbols mark the extrapolated spectrum from numerical computation. Different conformal multiplets are classified according to their degeneracy and tower structure. Dashed lines indicate the conformal tower generated from the lowest-dimension state within each multiplet (see main text).

### 4.2.3 Free/fixed-mixed b.c.

In this subsection, we present numerical results for boundary conditions connecting free or fixed b.c. with mixed b.c.s. Due to the $S_5$ permutation symmetry, the free-mixed b.c. leads to a single possible combination with respect to the four-state mixed b.c. The real parts of these boundary states are illustrated in Fig. 7. The extrapolated spectrum exhibits a conformal structure similar to other boundary fixed points, allowing the classification of different conformal multiplets. Within each conformal family, all states transform according to the same representation under the global symmetry, while their real parts differ by integers and their imaginary parts are nearly equal. This mixed b.c. breaks the original $S_5$ global symmetry of the model down to its $S_4$ subgroup. First, we identify the singlet identity operator and its corresponding conformal multiplets. Its first-level descendant vanishes, while the second- and third-level descendant fields are singly degenerate. Next, we identify two operators corresponding to $h_{3,1}$, which transform respectively as an $S_4$ singlet and as the three-dimensional standard representation of $S_4$. Their primary fields have scaling dimensions around $1 + 0.3063\,i$, and a null state appears at the third level. Following this, we find an eightfold-degenerate state with scaling dimension near $h_{5,1}$. Consequently, the corresponding annulus partition function can be written as (67):

$$Z_{\text{free,ABCD}} = \chi_{1,1} + 4\chi_{3,1} + 8\chi_{5,1} + \cdots. \tag{67}$$

Next, we consider fixed b.c. combined with the four-state mixed boundary condition, which results in two inequivalent boundary states, see Fig. 7. In both cases, the conformal multiplet of the leading singlet operator has a null state at the second level. After shifting the ground state energy to $h_{2,1}$, the next operator, which is threefold degenerate, has a scaling

dimension very close to $h_{4,1}$. Furthermore, under the A-BCDE mixed boundary condition, we observe a fivefold-degenerate operator whose representation is close to $\chi_{6,1}$. Accordingly, we numerically determine the following annulus partition functions:

$$Z_{A,ABCD} = \chi_{2,1} + 3\chi_{4,1} + \cdots, \tag{68}$$

$$Z_{A,BCDE} = \chi_{2,1} + 3\chi_{4,1} + 5\chi_{6,1} + \cdots. \tag{69}$$

It is worth noting that, although these cases can in principle be included via analytic continuation from the 2BTL annulus partition function (for the four-state mixed boundary condition, the blob parameter $r$ corresponds to 1, consistent with the original construction), the resulting Virasoro characters differ significantly from those observed numerically. One possibility is that, in the $Q = 5$ Potts CFT, these mixed boundary conditions do not correspond directly to any boundary fixed point. In this case, the 2BTL construction would need to be re-examined to check for potential inconsistencies. Another possibility is that, in this regime, the annulus partition functions of the Potts model and the loop model receive additional corrections.

## 4.3 Duality between boundary conditions

Lastly, we discuss some properties from the perspective of Kramers-Wannier duality, which is an important non-invertible symmetry of the Potts model. For a periodic chain, the critical bulk Potts model is invariant under this duality transformation, as discussed in Sec. 3. For open boundary conditions, this transformation can be explicitly written as

$$\begin{aligned}
\sigma'_{i+\frac{1}{2}} &= \Pi^i_{j=1} \tau_j, \\
\tau'_{i+\frac{1}{2}} &= \sigma^\dagger_i \sigma_{i+1}.
\end{aligned} \tag{70}$$

However, under this duality transformation, the boundary Hamiltonian is mapped to different boundary conditions. We illustrate this using the free-free b.c. case. When $g_L = g_R = 0$, the model in Eq. (49) is transformed into a new Hamiltonian:

$$\begin{aligned}
H'(J, h, \lambda) = &-J \sum_{i=1}^{L-1} \sum_{k=1}^{Q-1} \tau'^k_{i+\frac{1}{2}} - h \sum_{i=2}^{L} \sum_{k=1}^{Q-1} \left( \sigma'^\dagger_{i-\frac{1}{2}} \sigma'_{i+\frac{1}{2}} \right)^k \\
&+ \lambda \sum_{i=2}^{L-1} \sum_{k_1,k_2=1}^{Q-1} \left[ \left( \sigma'^\dagger_{i-\frac{1}{2}} \sigma'_{i+\frac{1}{2}} \right)^{k_1} \tau'^{k_2}_{i+\frac{1}{2}} + \tau'^{k_2}_{i+\frac{1}{2}} \left( \sigma'^\dagger_{i-\frac{1}{2}} \sigma'_{i+\frac{1}{2}} \right)^{k_1} \right] \\
&+ \lambda \sum_{i=1}^{L-1} \sum_{k_1,k_2=1}^{Q-1} \left[ (\sigma'^\dagger_{i+\frac{1}{2}} \sigma'_{i+\frac{3}{2}})^{k_1} \tau'^{k_2}_{i+\frac{1}{2}} + \tau'^{k_2}_{i+\frac{1}{2}} (\sigma'^\dagger_{i+\frac{1}{2}} \sigma'_{i+\frac{3}{2}})^{k_1} \right] \\
&-h \sum_{k=1}^{Q-1} \sigma'^k_{\frac{3}{2}} + \lambda \sum_{k_1,k_2=1}^{Q-1} (\sigma'^{k_1}_{\frac{3}{2}} \tau'^{k_2}_{\frac{3}{2}} + \tau'^{k_2}_{\frac{3}{2}} \sigma'^{k_1}_{\frac{3}{2}}).
\end{aligned} \tag{71}$$

Here, the original sites $\{1, 2, \cdots, L\}$ are relabeled as $\{\frac{3}{2}, \frac{5}{2}, \cdots, L+\frac{1}{2}\}$ in the transformed Hamiltonian. At the boundary critical point $J_c = h_c = 1$ and $\lambda_c \approx 0.079 + 0.060i$, the transformed model $H'$ contains transverse field, longitudinal field, and transverse-longitudinal interaction terms at its left end ($i = \frac{3}{2}$), while no on-site interaction exists at the right end ($i = L + \frac{1}{2}$). Since $|\lambda| \ll |J| = |h|$, the transverse-longitudinal interaction is negligible compared to the other terms. The longitudinal field tends to polarize the left end into the A direction, so this boundary is expected to renormalize into a fixed b.c. Meanwhile, the right-end spin is free to take any of the $Q$ states, corresponding to a sum over fixed b.c. with A, B, C, D, and E states, commonly referred to as the wired b.c. [25, 67].

This demonstrates that the free b.c. is dual to the wired b.c., and the corresponding partition function satisfies

$$Z_{\text{free,free}} = Q\left(Z_{A,A} + Z_{A,B} + Z_{A,C} + Z_{A,D} + Z_{A,E}\right) = Q\left(Z_{A,A} + 4Z_{A,B}\right). \tag{72}$$

Up to $\text{Re}(h) \leq 5$, we verify numerically that Eq. (72) holds using Eqs. (56), (60), and (61). This confirms the duality between free and wired b.c.s within our computations.

Furthermore, new boundary conditions can be constructed by fusing topological defect lines (TDLs), as studied systematically in Ref. [77] for rational CFTs. The duality transformation here can be understood in this framework, where the TDL corresponds to the Kramers-Wannier duality acting as a non-invertible symmetry [78,79] of the Potts model. Consequently, the duality between different boundary conditions is realized by fusing the duality defect line. While most previous works focus on fusing TDLs in real CFTs [77,80–83], here we extend these duality relations to complex CFTs, originating from the analytic continuation of the Potts CFT with $Q > 4$.

## 5 Summary, discussion and outlook

In this work, we have numerically investigated the boundary critical behavior of a non-Hermitian 5-state Potts spin chain, which exhibits complex conformal invariance in the bulk. We proposed a set of potential conformally invariant boundary conditions, inherited from the known $Q \leq 4$ Potts BCFT, and examined them using the characteristic conformal tower structures of operator spectra. We further extracted approximate conformal dimensions of boundary operators associated with irreducible representations of the Virasoro algebra. These results suggest that conformally invariant boundary fixed points persist even in the complex CFT regime.

Several directions for future research are apparent. First, it is important to develop theoretical methods to determine the operator content for general complex boundary fixed points. In many cases considered here, the boundary conformal spectra can be understood within the 2BTL algebra framework [21]. However, certain cases show discrepancies between the results obtained via direct analytic continuation of the 2BTL construction and our numerical findings. A natural direction for further study is therefore to investigate the algebraic structure of the boundary loop model for $Q > 4$ and to understand the origin of these discrepancies. Another interesting avenue is to interpret these boundary spectra from the perspective of logarithmic BCFTs, for which simple examples have been studied previously [84–90].

Second, once conformal boundary conditions are identified, one can in principle compute bulk-boundary OPE coefficients from wavefunction overlaps and thereby extract additional conformal data [91, 92]. Interpreting these data in the context of complex CFT requires careful understanding of the emergent boundary criticality. Such analyses could inform not only computations of complex boundary criticality, but also studies of boundary pseudo-critical behavior, in which a conformal boundary condition is applied to a bulk exhibiting pseudo-criticality due to nearby complex fixed points.

Third, the free, fixed, and mixed boundary conditions studied here may represent only a subset of all possible conformal boundary conditions, since the bulk CFT contains infinitely many primary fields. It remains important to explore additional boundary fixed points, such as those analogous to the 'new' boundary conditions for the 3- or 4-state Potts models [67], or by investigating the loop model with different weights for loops touching the boundary [21,36]. Another approach is to introduce relevant perturbations to a subsystem, which are expected to correspond to certain conformal boundary conditions of the unperturbed bulk [93–101]. This strategy also enables studies of boundary RG flows via the $g$-function [102].

For instance, in our model one could introduce a small perturbation at the boundary under free boundary conditions and monitor the evolution of the spectrum as the system size increases. This approach may allow direct observation of RG flows between boundary fixed points in the complex CFT. While a $c$-theorem for bulk complex fixed points has not yet been established [103], extending the $g$-function to complex boundary fixed points remains an open problem.

Finally, as discussed in Section 4.3, boundary states can also be constructed by fusing topological defect lines (TDLs), a technique well established in rational CFTs [77, 80–83]. Extending this perspective to complex CFTs may provide further insights. Moreover, lattice realizations of TDLs [104–108] in the non-Hermitian 5-state Potts model offer a promising avenue for gaining a deeper understanding of the defect phenomena.

## Acknowledgments

We thank Han Ma and Yin-Chen He for collaboration on the previous project. We thank M. Oshikawa, Chenjie Wang and Xueda Wen for helpful discussion. We are especially grateful to Jesper Jacobsen for detailed and insightful comments on boundary loop models and blob algebra, which greatly helped us to clarify the connection to existing analytical results.

**Funding information** This work was supported by the National Natural Science Foundation of China No.12474144, the National Key R&D Program No.2022YFA1402204 and the foundation of Westlake University (YT, QL, WZ).

## A More numerical results

In this section, more numerical data of the boundary operator content for fixed, free-mixed and fixed-mixed b.c.s are presented. Since the descendant fields are related to the primaries with integer-deviation, we only exhibit the scaling dimensions of boundary primaries in the following. In addition, the Kac index for the low-lying fields has been identified within the main text. Thus we compare the theoretical prediction and numerical results in Tab. 3 and Tab. 4.

Table 3: The scaling dimensions of boundary primaries for fixed boundary conditions. Different boundary conditions are listed in the first column. The second column presents the representation of different boundary operators $\phi_{r,s}$ specified by Kac index $r,s$. Then the theoretical prediction and numerical results after extrapolation are presented within the last two column. For the A-B and free-A cases, the identity multiplet is not included within the boundary operator content and we shift the whole spectrum by the lowest scaling dimensions. For each boundary condition, the spectrum is normalized according to the integer spacing within the same conformal multiplet, and then shifted so that the ground state corresponds to the scaling dimension of the corresponding Virasoro primary field. Therefore, the lowest levels shown here are exact by construction. A more detailed explanation of the normalization procedure is provided in the main text.

| Boundary Condition | Operators | theoretical | numerical |
|---|---|---|---|
| A-A | $\phi_{1,1}$ | 0 | 0 |
| | $\phi_{5,1}$ | $4 + 0.9190i$ | $4.0216 + 0.9241i$ |
| A-B | $\phi_{3,1}$ | $1 + 0.3063i$ | $1 + 0.3063i$ |
| | $\phi_{5,1}$ | $4 + 0.9190i$ | $4.0063 + 0.9233i$ |
| free-A | $\phi_{2,1}$ | $0.25 + 0.1149i$ | $0.25 + 0.1149i$ |
| | $\phi_{4,1}$ | $2.25 + 0.5744i$ | $2.2542 + 0.5768i$ |

Table 4: The scaling dimensions of boundary conformal primaries for free-mixed boundary conditions.

| Boundary Condition | Operators | theoretical | numerical |
|---|---|---|---|
| free-ABCD | $\phi_{1,1}$ | 0 | 0 |
| | $\phi_{3,1}$ | $1 + 0.3063i$ | $0.8766 + 0.2817i$ |
| | $\phi_{5,1}$ | $4 + 0.9190i$ | $3.7667 + 0.8687i$ |
| A-ABCD | $\phi_{2,1}$ | $0.25 + 0.1149i$ | $0.25 + 0.1149i$ |
| | $\phi_{4,1}$ | $2.25 + 0.5744i$ | $2.0894 + 0.5261i$ |
| A-BCDE | $\phi_{2,1}$ | $0.25 + 0.1149i$ | $0.25 + 0.1149i$ |
| | $\phi_{4,1}$ | $2.25 + 0.5744i$ | $1.7175 + 0.5238i$ |
| | $\phi_{6,1}$ | $6.25 + 1.3403i$ | $5.2568 + 1.1818i$ |

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
