# Peer review of "Boundary Criticality of Complex Conformal Field Theory: A Case Study in the Non-Hermitian 5-State Potts Model"

_SciPost Physics, doi:SciPost Phys. 19, 164 (2025)_

## Round 1 · Referee Report · Anonymous (Referee 1) · 2025-1-20

Report

This paper using exact diagonalization to study a Hamiltonian associated with the 5-state Potts model, which is non-Hermitian. The results for the spectrum of the theory match exact predictions for various boundary conditions, and confirm these previous predictions that the fixed point is a complex CFT.

The paper seems correct and a nice addition to the literature. My main issue is that there are many grammatical mistakes throughout the paper, which make it somewhat difficult to read. I would advice the authors to use ChatGPT or something similar to proofread the text for grammar, before resubmitting.

Recommendation

Ask for minor revision

  • validity: -
  • significance: -
  • originality: -
  • clarity: -
  • formatting: -
  • grammar: -

Author:  Yin Tang  on 2025-11-11  [id 6004]

(in reply to Report 1 on 2025-01-20)
Category:
answer to question

Referee comment:

This paper uses exact diagonalization to study a Hamiltonian associated with the 5-state Potts model, which is non-Hermitian. The results for the spectrum of the theory match exact predictions for various boundary conditions, and confirm these previous predictions that the fixed point is a complex CFT.
The paper seems correct and a nice addition to the literature. My main issue is that there are many grammatical mistakes throughout the paper, which make it somewhat difficult to read. I would advise the authors to use ChatGPT or something similar to proofread the text for grammar, before resubmitting.

Reply:
We thank the referee for the positive evaluation of our work and for the constructive comment. We have carefully revised the entire manuscript to improve the English presentation and readability. In particular, we have thoroughly checked the grammar, sentence structure, and overall clarity of the text through both manual editing and AI-assisted proofreading tools, ensuring that the revised version is substantially clearer and more fluent.

---

## Round 1 · Referee Report · Anonymous (Referee 2) · 2025-3-23

Report

In this paper, the authors study the boundary critical phenomena in the quantum non-Hermitian 5-state Potts chain, which is described in terms of a complex Conformal Field Theory (CFT). The complex CFT is a novel area of study and of significant current interest especially in its relevance for the pseudo-critical behavior of the standard Hermitian 5-state Potts chain (equivalent to the standard 2-dimensional 5-state Potts model in classical statistical mechanics). The boundary "complex CFT" is an unexplored area and this paper is a significant first step into that direction. The combination of numerical and theoretical analyses lead to beautiful results extending the known boundary (non-complex) CFTs. This is an important piece of work which I can strongly recommend for publication in SciPost. However, I would like to ask the authors to revise the paper on the following points before publication. - In studying the non-free boundary conditions, the authors introduce the boundary fields $h_{L,R}$. However, the description "In all cases, we take $h_L$ and $h_R$ to be a large number to ensure the long-wavelength limit of this model could flow to corresponding boundary fixed points." in the paper is rather vague. For the reproducibility of the numerical results, the authors should provide more details (actual values of $h_{L,R}$ used in the actual calculations. You can send $h_{L,R}$ to (positive) infinity, so that the boundary spins are retricted to the subset of 5 states --- this is used in many other works and maybe also in this paper but please clarify as much as practically possible. Of course, changing the values of $h_{L,R}$ to observe the boundary Renormalization Group flow would be interesting, but perhaps it goes beyond the scope of the paper. - In any case, if $h_{L,R}$ are negative you will see a different boundary condition, so perhaps it should be declared that $h_{L,R} \geq 0$. - I find it unfortunate that the same symbol $h$ is used for the conformal dimension $h_{r,s}$, the bulk field $h$, and the boundary fields $h_{L,R}$. Most of the time it is obvious from the context, but it is still somewhat confusing (e.g. when you say $\mathrm{Re}{(h)} \leq 5$.) Maybe you can consider using different symbols. - The analysis of the scaling dimensions through the rescaled gap as in Eq. (27) is reasonable. The "numerical" value of 2 in the second row is exact because of the way of the analysis; this is an obvious consequence of Eq. (27) but it would still be nice to comment that in Table 2. (also applies Table 3). - Table 3: "the whole spectrum is shifted by the lowest scaling dimension within each b.c."--- it would be nice to describe more details explicitly (although I can guess roughly what was done). - By comparing the actual finite-size gap and the expected conformal dimension, one can estimate the "spin-wave velocity" for the critical non-Hermitian 5-state Potts chain. While its value is not relevant for the main objective of the present study, it would be nice to provide the value as a reference. - I agree with Referee 1 that the English of this paper should be improved (especially since the content is great!). Just for example, in page 3, "the stability under RG flows is not aware" --- "aware" is used for example "he is not aware of the fact that..." and "stability" cannot be aware of anything. (Perhaps you can say "... is not known".) "If exist,..." perhaps it should be "If they exist,..." There are many more issues throughout the paper. While the paper is largely comprehensible, there is much room for improvement for better readability.

Recommendation

Ask for minor revision

  • validity: -
  • significance: -
  • originality: -
  • clarity: -
  • formatting: -
  • grammar: -

Author:  Yin Tang  on 2025-11-11  [id 6005]

(in reply to Report 2 on 2025-03-23)

Referee comment:

In this paper, the authors study the boundary critical phenomena in the quantum non-Hermitian 5-state Potts chain, which is described in terms of a complex Conformal Field Theory (CFT). The complex CFT is a novel area of study and of significant current interest especially in its relevance for the pseudo-critical behavior of the standard Hermitian 5-state Potts chain (equivalent to the standard 2-dimensional 5-state Potts model in classical statistical mechanics). The boundary "complex CFT" is an unexplored area and this paper is a significant first step into that direction. The combination of numerical and theoretical analyses lead to beautiful results extending the known boundary (non-complex) CFTs. This is an important piece of work which I can strongly recommend for publication in SciPost.

Reply:
We thank the referee for the positive assessment and for recommending our work for publication.

Referee comment:

However, I would like to ask the authors to revise the paper on the following points before publication.
- In studying the non-free boundary conditions, the authors introduce the boundary fields $h_{L,R}$. However, the description "In all cases, we take $h_L$ and $h_R$ to be a large number to ensure the long-wavelength limit of this model could flow to corresponding boundary fixed points." in the paper is rather vague. For the reproducibility of the numerical results, the authors should provide more details (actual values of $h_{L,R}$ used in the actual calculations. You can send $h_{L,R}$ to (positive) infinity, so that the boundary spins are restricted to the subset of 5 states — this is used in many other works and maybe also in this paper but please clarify as much as practically possible. Of course, changing the values of $h_{L,R}$ to observe the boundary Renormalization Group flow would be interesting, but perhaps it goes beyond the scope of the paper.
- In any case, if $h_{L,R}$ are negative you will see a different boundary condition, so perhaps it should be declared that $h_{L,R} \geq 0$.

Reply:
We thank the referee for this helpful comment. In the revised manuscript, we have clarified the implementation of the boundary fields ($g_{L,R}$) in the main text. Specifically, we now state that in all numerical calculations we take $g_L = g_R = 20$, which is at least one order of magnitude larger than the coupling constants in the bulk Hamiltonian. This choice ensures that the imposed boundary conditions flow to the corresponding boundary fixed points in the low-energy limit. We also specify that $g_{L,R}$ must be taken to be positive, as negative values would correspond to different boundary fixed points. In addition, we have added a short discussion in the Discussion section on how varying $g_{L,R}$ could be used to study the boundary RG flows between different fixed points, as suggested by the referee.

Referee comment:

I find it unfortunate that the same symbol h is used for the conformal dimension $h_{r,s}$, the bulk field $h$, and the boundary fields $h_{L,R}$. Most of the time it is obvious from the context, but it is still somewhat confusing (e.g. when you say $\text{Re}(h)\leq 5$.) Maybe you can consider using different symbols.

Reply:
We thank the referee for pointing out this notational ambiguity. In the revised manuscript, we have changed the notation of the boundary interaction strength from $h_{L,R}$ to $g_{L,R}$ to avoid confusion. The symbols $h_{r,s}$ and $h$ referring to the scaling dimensions of Virasoro representation and boundary fields are kept unchanged. In addition, around the sentence mentioning ($\text{Re}(h)\leq 5$), we have added an explicit clarification.

Referee comment:

The analysis of the scaling dimensions through the rescaled gap as in Eq. (27) is reasonable. The "numerical" value of 2 in the second row is exact because of the way of the analysis; this is an obvious consequence of Eq. (27) but it would still be nice to comment that in Table 2. (also applies Table 3).
Table 3: "the whole spectrum is shifted by the lowest scaling dimension within each b.c." — it would be nice to describe more details explicitly (although I can guess roughly what was done).

Reply:
We thank the referee for this helpful suggestion. Following the advice, we have clarified these points directly in the captions of Tables 2 and 3. In Table 2, we now state that in the numerical analysis, the two lowest non-degenerate states are normalized to have scaling dimensions 0 and 2, respectively, in order to rescale the entire spectrum. These two states correspond to the identity operator $\phi_{1,1}$ and its first non-null descendant $L_{-2}\phi_{1,1}$. Consequently, their scaling dimensions are exactly 0 and 2 in the numerical results. In Table 3, we explain that the spectrum is normalized according to the integer spacing within the same conformal multiplet, and then shifted so that the ground state corresponds to the scaling dimension of the corresponding Virasoro primary field. Therefore, the lowest levels in the table are exact by construction. A more detailed description of the procedure has been included in the main text.

Referee comment:

By comparing the actual finite-size gap and the expected conformal dimension, one can estimate the "spin-wave velocity" for the critical non-Hermitian 5-state Potts chain. While its value is not relevant for the main objective of the present study, it would be nice to provide the value as a reference.

Reply:
We thank the referee for this useful suggestion. In the revised manuscript, we have added the extracted complex “speed of light” (or spin-wave velocity) in the caption of Fig.~2 for reference. Specifically, we now state that $v = \frac{A}{2\pi} \simeq 2.7205 - 0.6906i$ (fitting data from $L=6$ to $L=11$), which is very close to the periodic boundary condition case, where $v_{\text{PBC}} \simeq 2.8810 - 0.7091i$ (fitting data from $L=8$ to $L=13$).

Referee comment:

I agree with Referee 1 that the English of this paper should be improved (especially since the content is great!). Just for example, in page 3, "the stability under RG flows is not aware" — "aware" is used for example "he is not aware of the fact that..." and "stability" cannot be aware of anything. (Perhaps you can say "... is not known".) "If exist,..." perhaps it should be "If they exist,..." There are many more issues throughout the paper. While the paper is largely comprehensible, there is much room for improvement for better readability.

Reply:
We thank the referee for the constructive comment regarding the English writing. We have corrected the specific examples mentioned (“is not aware” → “is not known and fully understood”, “If exist” → “If these fixed points exist”) and carefully proofread the entire manuscript to improve grammar, clarity, and readability throughout.

---

## Round 1 · Referee Report · Jesper Lykke Jacobsen (Referee 3) · 2025-4-6

Strengths

1-First numerical study of boundary conditions in a complex CFT

Weaknesses

1-Does not connect to existing results about boundary conditions in non-unitary CFT.
2-Lacks precision about representation-theoretical aspects.

Report

It was recently argued that conformally invariant critical behaviour can exist in the Potts model with $Q > 4$ states, provided the coupling constant of the field theory (Coulomb gas) is analytically continued to complex values. This prediction was followed by two numerical studies of specific models, where this scenario is realised. The first one (ref. [25]) studied a loop model on the triangular lattice, for which the loop fugacity can encode any complex value of Q. The second one (ref. [26]) studied a non-hermitian 1+1D spin chain, with integer $Q=5$. In both cases it was found that CFT properties such as the central charge and bulk critical exponents are in accord with the scenario of analytic continuation.

The present manuscript continues the study of the $Q=5$ spin chain, but now focusing on boundary critical behaviour. The results indicate that also the boundary-CFT properties are in agreement with analytic continuation. Unfortunately, in a number of cases the authors themselves seem unable to fully reach this conclusion, as they appear to be ignorant of a large body of existing results for non-unitary models that were previously studied for real values of $Q < 4$. This flaws the manuscript in two ways. First, it leads in a number of cases the authors to claim novelty for things which are in fact well known. Second, it often prevents the authors from reaching clear conclusions, even when they could have done so.

The problem becomes conspicuous already in the introduction, where the authors state that "the investigation of irrational cases [of boundary CFT] is difficult (...), [with results being] limited to numerical calculations". Fortunately this is not so. The loop models of Potts and $O(n)$ type, which beautifully realise the irrational CFT with real $c < 1$, and which are closely related to the Coulomb-gas approach in the bulk case, have been thoroughly studied also in the boundary case. For instance, the study of a set of boundary conditions that amount to allowing a lesser number of states, $Q_1 < Q$, for the spins along one boundary was initiated by Jacobsen and Saleur in Nucl. Phys. B 788, 137 (2008), then further studied and extended by the same authors and their collaborators in a long series of papers over the following years. These boundary conditions contain all those being studied in this manuscript as special cases.

Section 2 contains a review of some well-known features of bulk and boundary CFT. The focus is to a large extent on minimal models. The authors present, misleadingly in view of the comments made above, the complex CFT studied here as the first example of non-unitary CFT. But in fact, most features of complex CFT are already present in the non-unitary CFT with $Q < 4$. Other details need correction, too. After eq. (5) it is stated that the "formalism was originally proposed for integer $Q \le 4$", whereas it is in fact for integer $m$, viz. for the minimal models. After eq. (6) the "higher represetations" alluded to are more precisely representations of the global symmetry $S_Q$. The discussion about Cardy conditions and Ishibashi states applies, as stated, only to rational CFT, but this is not mentioned. Towards the end of section 2, the statement that there are "little [few] known results for irrational BCFT" is again despairing.

In section 3 the spin chain being studied is defined. It would have been nice to know more about the boundary fields, in particular how should they be chosen in order to respect quantum-group symmetry? Shortly after eq. (25) the authors suddenly mention that the free and fixed boundary conditions that they study are "known as blob bcs". This terminology is derived from the literature related to the Jacobsen-Saleur paper cited above, via an earlier paper by Martin and Saleur, Lett. Math. Phys. 30, 189 (1994), so it is surprising that this is brought up without the proper context (only a number of much later and tangentially related numerical papers are cited).

In section 4 the authors finally arrive at their numerical results. The authors should have stated up to which size the computations have been made (judging from the figures it is $L=11$). Some of the representation-theoretical statements should be made more precise, for instance in the caption to figure 3 the "lowest $S_5$ vector operator" is presumably the one associated with the Young diagram $(4,1)$, but it is not clear what is meant by "the lowest higher-representation field".

In the discussion about free-free boundary conditions (section 4.2.1), it is not clear what is meant by "the non-unitary nature of the hidden fixed points". The first displayed formula in that section gives the decomposition of the annulus partition function in terms of characters, with coefficients $1, 4, 11$. It is a standard exercise of Temperley-Lieb representation theory that these coefficients are the even-order Chebyshev polynomials of the second kind, of which the first are $1$, $Q-1 = 4$, $Q^2 - 3Q + 1 = 11$ indeed. This can be found in Saleur and Bauer, Nucl. Phys. B 320, 591 (1989).

In the next section about fixed boundary conditions (section 4.2.2) the results can also be explained in terms of existing analytical results. For instance, the fact that the free/fixed boundary condition changing operator is $\phi_{2,1}$ was established by Cardy in J. Phys. A 25, L201 (1992) and put in a more general setting in section 3.2 of the paper by Jacobsen and Saleur cited above. Moreover, the expansion coefficients of $Z_{A,A}$ in eq. (31) can be inferred from eq. (3.19) in Jacobsen and Saleur, J. Stat. Mech. (2008) P01021 (take the sum of the two equations). The first non-trivial coefficient is $Q - 2 = 3$ as observed in eq. (31), and the next coefficient (not provided there) should be $Q^2 - 4Q + 2 = 7$. The remaining results can be established from the two-boundary Temperley-Lieb model, which was further studied e.g. in Dubail, Jacobsen and Saleur, Nucl. Phys. B 813, 430 (2009).

The following section on free/fixed-mixed boundary conditions (section 4.2.3) contains a number of vague (and hence rather incomprehensible) statements. For instance, what is meant by "it is difficult to align each multiplet with corresponding Virasoro characters"? Same thing about the mentioning of the Kac indices being allowed to take fractional values: it is not written what those fractional values are. It seems that the authors' findings about fixed/mixed boundary conditions are inconclusive, but results are available in the papers on the two-boundary TL model cited above. A final vague statement is that "the symmetry of most configurations is much smaller than...".

In section 4.3 about duality, it should be mentioned that the sum over fixed boundary conditions is usually called wired boundary conditions. There is a global factor of $Q$ missing in the displayed formula. It is wrong to state that there is a duality between free and fixed boundary conditions: the correct statement is that there is a duality between free and wired. And the final claim that "here for the first time we consider such duality relations" is at best misleading: all of these results are well understood in the context of non-unitary models with real values of Q (see the papers cited above), and so the extension to complex $Q$ is hardly surprising.

In the same vein, the concluding section 5 is too optimistic about what the authors have achieved and oblivious about the existing literature.

In conclusion, this manuscript most definitely cannot be accepted in its present form, although it does open the interesting subject of complex boundary CFT. It is possible that a thorough revision could be reconsidered for publication in SciPost Core, but in my opinion not in the flagship journal SciPost Phys.

Requested changes

See main report.

Recommendation

Ask for major revision

  • validity: high
  • significance: ok
  • originality: ok
  • clarity: good
  • formatting: excellent
  • grammar: acceptable

Author:  Yin Tang  on 2025-11-11  [id 6006]

(in reply to Report 3 by Jesper Lykke Jacobsen on 2025-04-06)
Category:
answer to question
correction

Referee comment:

It was recently argued that conformally invariant critical behaviour can exist in the Potts model with $Q>4$ states, provided the coupling constant of the field theory (Coulomb gas) is analytically continued to complex values. This prediction was followed by two numerical studies of specific models, where this scenario is realised. The first one (ref. [25]) studied a loop model on the triangular lattice, for which the loop fugacity can encode any complex value of $Q$. The second one (ref. [26]) studied a non-hermitian 1+1D spin chain, with integer $Q=5$. In both cases it was found that CFT properties such as the central charge and bulk critical exponents are in accord with the scenario of analytic continuation.
The present manuscript continues the study of the $Q=5$ spin chain, but now focusing on boundary critical behaviour. The results indicate that also the boundary-CFT properties are in agreement with analytic continuation. Unfortunately, in a number of cases the authors themselves seem unable to fully reach this conclusion, as they appear to be ignorant of a large body of existing results for non-unitary models that were previously studied for real values of $Q<4$. This flaws the manuscript in two ways. First, it leads in a number of cases the authors to claim novelty for things which are in fact well known. Second, it often prevents the authors from reaching clear conclusions, even when they could have done so.

Reply:
We thank the referee for the careful and detailed review of our manuscript, and for pointing out several important related works that we were previously unaware of. These references have greatly helped us to place our results in the proper context and to improve our understanding of the numerical findings. We have incorporated the relevant literature and corresponding discussions in the revised version, which has significantly clarified the connections between our work and existing studies on non-unitary models with real $Q<4$.

Referee comment:

The problem becomes conspicuous already in the introduction, where the authors state that "the investigation of irrational cases [of boundary CFT] is difficult (...), [with results being] limited to numerical calculations". Fortunately this is not so. The loop models of Potts and $O(n)$ type, which beautifully realise the irrational CFT with real $c<1$, and which are closely related to the Coulomb-gas approach in the bulk case, have been thoroughly studied also in the boundary case. For instance, the study of a set of boundary conditions that amount to allowing a lesser number of states, $Q_1<Q$, for the spins along one boundary was initiated by Jacobsen and Saleur in Nucl. Phys. B 788, 137 (2008), then further studied and extended by the same authors and their collaborators in a long series of papers over the following years. These boundary conditions contain all those being studied in this manuscript as special cases.

Reply:
We thank the referee for this helpful remark and for pointing out the important body of work on boundary loop models. In the revised version, we have clarified our statement in the introduction. Our original sentence referred to generic irrational boundary CFTs, for which analytical approaches are scarce. We now explicitly acknowledge that in special cases, such as the two-dimensional Potts and $O(n)$ loop models, the presence of an enlarged algebra (the Temperley–Lieb algebra and its boundary extensions) provides an analytic handle on the boundary conformal spectra, even for non-unitary irrational theories. The statement in the revised manuscript has been modified accordingly.

Referee comment:

Section 2 contains a review of some well-known features of bulk and boundary CFT. The focus is to a large extent on minimal models. The authors present, misleadingly in view of the comments made above, the complex CFT studied here as the first example of non-unitary CFT. But in fact, most features of complex CFT are already present in the non-unitary CFT with $Q<4$. Other details need correction, too. After eq. (5) it is stated that the "formalism was originally proposed for integer $Q \leq 4$", whereas it is in fact for integer $m$, viz. for the minimal models. After eq. (6) the "higher represetations" alluded to are more precisely representations of the global symmetry $S_Q$. The discussion about Cardy conditions and Ishibashi states applies, as stated, only to rational CFT, but this is not mentioned. Towards the end of section 2, the statement that there are "little [few] known results for irrational BCFT" is again despairing.

Reply:
We thank the referee for the detailed and constructive comments regarding Section 2. We have carefully revised this section to address all the raised points. Specifically:
We clarified that the construction of Ishibashi and Cardy states applies only to rational CFTs, and that the systematic construction of boundary states in generic irrational cases remains unknown.
We corrected the statement after eq. (5), noting that the formalism was originally proposed for integer $m$ rather than for integer $Q\le4$.
We also clarified that the “representations” mentioned after eq. (6) refer more precisely to representations of the global symmetry group $S_Q$.
Finally, we have added a short review at the end of Section 2 summarizing how, in the irrational Potts BCFT with $Q<4$, the boundary Temperley–Lieb algebra can be used to construct annulus partition functions and to analytically continue boundary spectra.

Referee comment:

In section 3 the spin chain being studied is defined. It would have been nice to know more about the boundary fields, in particular how should they be chosen in order to respect quantum-group symmetry? Shortly after eq. (25) the authors suddenly mention that the free and fixed boundary conditions that they study are "known as blob bcs". This terminology is derived from the literature related to the Jacobsen-Saleur paper cited above, via an earlier paper by Martin and Saleur, Lett. Math. Phys. 30, 189 (1994), so it is surprising that this is brought up without the proper context (only a number of much later and tangentially related numerical papers are cited).

Reply:
We thank the referee for pointing this out. We have now added the proper reference at the place where the “blob boundary conditions” are introduced. In addition, we have briefly reviewed the relevant background and main results of the blob boundary construction in Sections 2.3 and 3.3.

Referee comment:

In section 4 the authors finally arrive at their numerical results. The authors should have stated up to which size the computations have been made (judging from the figures it is $L=11$). Some of the representation-theoretical statements should be made more precise, for instance in the caption to figure 3 the "lowest $S_5$ vector operator" is presumably the one associated with the Young diagram (4,1), but it is not clear what is meant by "the lowest higher-representation field".

Reply:
We thank the referee for these useful comments. We have clarified at the beginning of Section 4.1 that our numerical calculations were performed for system sizes $L = 6-11$. We have also revised the caption of Fig.3 and related text to make the representation-theoretical statements more precise: the phrase “lowest $S_5$ vector operator” has been replaced with “the most relevant $S_5$ 4-dimensional standard representation operators,” corresponding to the Young diagram (4,1).

Referee comment:

In the discussion about free-free boundary conditions (section 4.2.1), it is not clear what is meant by "the non-unitary nature of the hidden fixed points". The first displayed formula in that section gives the decomposition of the annulus partition function in terms of characters, with coefficients 1,4,11. It is a standard exercise of Temperley-Lieb representation theory that these coefficients are the even-order Chebyshev polynomials of the second kind, of which the first are $1$, $Q-1=4$, $Q^2-3Q+1=11$ indeed. This can be found in Saleur and Bauer, Nucl. Phys. B 320, 591 (1989).

Reply:
We thank the referee for this insightful remark. We have added a short discussion in Sec.~4.2.1 clarifying that the coefficients in Eq.~(50) can be derived from the two-boundary Temperley–Lieb annulus partition function with $r_1=r_2=r_{12}=1$, and that they correspond to the even-order Chebyshev polynomials of the second kind.

Referee comment:

In the next section about fixed boundary conditions (section 4.2.2) the results can also be explained in terms of existing analytical results. For instance, the fact that the free/fixed boundary condition changing operator is $\phi_{2,1}$ was established by Cardy in J. Phys. A 25, L201 (1992) and put in a more general setting in section 3.2 of the paper by Jacobsen and Saleur cited above. Moreover, the expansion coefficients of $Z_{A,A}$ in eq. (31) can be inferred from eq. (3.19) in Jacobsen and Saleur, J. Stat. Mech. (2008) P01021 (take the sum of the two equations). The first non-trivial coefficient is $Q-2=3$ as observed in eq. (31), and the next coefficient (not provided there) should be $Q^2-4Q+2=7$. The remaining results can be established from the two-boundary Temperley-Lieb model, which was further studied e.g. in Dubail, Jacobsen and Saleur, Nucl. Phys. B 813, 430 (2009).

Reply:
We thank the referee for pointing this out. We have added a short discussion at the end of Sec.~4.2.2 explaining how the results for fixed boundary conditions can be derived from the analytical formulations of the two-boundary Temperley–Lieb algebra.

Referee comment:

The following section on free/fixed-mixed boundary conditions (section 4.2.3) contains a number of vague (and hence rather incomprehensible) statements. For instance, what is meant by "it is difficult to align each multiplet with corresponding Virasoro characters"? Same thing about the mentioning of the Kac indices being allowed to take fractional values: it is not written what those fractional values are. It seems that the authors' findings about fixed/mixed boundary conditions are inconclusive, but results are available in the papers on the two-boundary TL model cited above. A final vague statement is that "the symmetry of most configurations is much smaller than...".

Reply:
We thank the referee for these valuable comments. We have reanalyzed the numerical results in Sec.~4.2.3 and rewritten this section accordingly.

Referee comment:

In section 4.3 about duality, it should be mentioned that the sum over fixed boundary conditions is usually called wired boundary conditions. There is a global factor of $Q$ missing in the displayed formula. It is wrong to state that there is a duality between free and fixed boundary conditions: the correct statement is that there is a duality between free and wired. And the final claim that "here for the first time we consider such duality relations" is at best misleading: all of these results are well understood in the context of non-unitary models with real values of Q (see the papers cited above), and so the extension to complex $Q$ is hardly surprising.

Reply:
We thank the referee for these valuable remarks. We have corrected the terminology throughout the text, now referring to the sum over fixed boundary conditions as the wired boundary condition. The missing global factor of $Q$ has been inserted in Eq.~(54), and the duality relation is now stated as being between free and wired boundary conditions. We have also revised the final paragraph of Section~4.3 to avoid the misleading claim and to clarify that our contribution is the extension of these duality relations to the case of complex CFT.

Referee comment:

In the same vein, the concluding section 5 is too optimistic about what the authors have achieved and oblivious about the existing literature.

Reply:
We appreciate the referee’s constructive remark. We have revised the concluding discussion in Sec.~5 to provide a more balanced summary of our results, with clearer connections to the existing literature and previous analytical works.

Referee comment:

In conclusion, this manuscript most definitely cannot be accepted in its present form, although it does open the interesting subject of complex boundary CFT. It is possible that a thorough revision could be reconsidered for publication in SciPost Core, but in my opinion not in the flagship journal SciPost Phys.

Reply:
We sincerely thank the referee for the comments and suggestions, which are helpful for us to improve the presentation and discussion. Now we believe the quality of this work has been significantly improved. We hope the referee could reconsider the revised manuscript for publication.

---

## Round 2 · Referee Report · Jesper Lykke Jacobsen (Referee 3) · 2025-11-11

Strengths

1-This provides what appears to be the first numerical study of boundary properties for the complex CFT of the 5-state Potts model.
2-The discussion how many of these results emerge from the analytic continuation of results on boundary loop models has been much improved with respect to the previous version.

Weaknesses

1-A few results are still unexplained and would require future work.

Report

The authors have very carefully taken into account my comments on the earlier version, leading to a substantial improvement of the paper. I can now recommend publication in SciPost Physics.

Requested changes

1-Given the very extensive changes and improvements that follow directly from the remarks made in my report on the first version of this manuscript (several pages of text have been added), I think it would be appropriate that the authors recognize this in the acknowledgments.

Recommendation

Publish (easily meets expectations and criteria for this Journal; among top 50%)

---

## Round 2 · List of Changes

1. As suggested by the first two referees, we have improved the overall readability of the manuscript by revising the English presentation throughout.

  2. As suggested by the second referee, we have clarified the implementation of the boundary fields by specifying that $g_L = g_R = 20$ and added related discussion on boundary RG flows in the Discussion section.

  3. As suggested by the second referee, we have clarified in Tables 2 and 3 how the normalization of the spectra is performed and explained why certain scaling dimensions are exact by construction.

  4. Following the second referee’s comment, we have added the extracted complex spin-wave velocity ($v = 2.7205 - 0.6906i$) in the caption of Fig. 2 for reference.

  5. As suggested by the third referee, we have clarified that the construction of Cardy and Ishibashi states applies only to rational BCFTs.

  6. As suggested by the third referee, we have revised the introduction to clarify that while generic irrational BCFTs are difficult to treat, cases with enlarged algebras (such as those in Potts and $O(n)$ loop models) admit exact algebraic treatments even for non-unitary irrational theories.

  7. As suggested by the third referee, we have corrected minor inaccuracies in Section~2.

  8. As suggested by the third referee, we have added a short review at the end of Section~2 summarizing the main results of algebraic construction of irrational Potts BCFTs with real $Q<4$, where the boundary Temperley--Lieb algebra allows one to build annulus partition functions analytically continued in $Q$.

  9. As suggested by the third referee, we have added proper references and contextual explanations when introducing the “blob boundary conditions.”

  10. Following the third referee’s suggestion, we now explicitly state the system size used in our exact diagonalization ($L = 6$$11$), and have clarified the representation-theoretical identification in Fig.~3. The “lowest $S_5$ vector operator” is now described as the most relevant operator in the 4-dimensional standard representation.

  11. As suggested by the third referee, we have added a paragraph in Sec.~4.2.1 explaining that the coefficients in Eq.~(50) could be derived from the two-boundary Temperley–Lieb annulus partition function with parameters $r_1=r_2=r_{12}=1$, and that they correspond to even-order Chebyshev polynomials of the second kind.

  12. We have added a new discussion at the end of Sec.~4.2.2 showing how the results for fixed boundary conditions can be derived from the analytic 2BTL framework.

  13. We have reanalyzed the numerical data and completely rewritten Sec.~4.2.3 to clarify the free/fixed–mixed boundary results, connecting them more explicitly with the analytic 2BTL predictions.

  14. As suggested by the third referee, we have corrected the terminology in Sec.~4.3, now referring to the “wired” boundary condition instead of the “fixed boundaries,” inserted the missing global factor of $Q$ in Eq.~(54), and revised the discussion of the duality.

  15. Finally, we have rewritten the concluding Section~5 to provide a more balanced summary of our contributions.

---

## Editorial Decision

published